# Striatal network modeling in Huntington's Disease

**Adam Ponzi** [1¤]*, **Scott J. Barton**[2], **Kendra D. Bunner**[2], **Claudia Rangel-Barajas**[2], **Emily S. Zhang**[2], **Benjamin R. Miller**[2], **George V. Rebec**[2☯‡], **James Kozloski**[1☯‡]

**1** IBM Research, Computational Biology Center, Thomas J. Watson Research Laboratories, Yorktown Heights, New York, United States of America, **2** Program in Neuroscience, Department of Psychological and Brain Sciences, Indiana University, Bloomington, Indiana, United States of America

☯ These authors contributed equally to this work.
¤ Current address: Institute of Biology, Otto-von-Guericke University, Magdeburg, Germany
‡ These authors are joint senior authors on this work.
* adamo.ponzi@ovgu.de

**Data Availability Statement:** The data are available at https://github.com/adampdp/plosCB-ponzi-HD.

**Funding:** Portions of each authors' work reported here were funded by CHDI. AP's work was also funded in part by NSF grant DMS-1555237. The

## Abstract

Medium spiny neurons (MSNs) comprise over 90% of cells in the striatum. *In vivo* MSNs display coherent burst firing cell assembly activity patterns, even though isolated MSNs do not burst fire intrinsically. This activity is important for the learning and execution of action sequences and is characteristically dysregulated in Huntington's Disease (HD). However, how dysregulation is caused by the various neural pathologies affecting MSNs in HD is unknown. Previous modeling work using simple cell models has shown that cell assembly activity patterns can emerge as a result of MSN inhibitory network interactions. Here, by directly estimating MSN network model parameters from single unit spiking data, we show that a network composed of much more physiologically detailed MSNs provides an excellent quantitative fit to wild type (WT) mouse spiking data, but only when network parameters are appropriate for the striatum. We find the WT MSN network is situated in a regime close to a transition from stable to strongly fluctuating network dynamics. This regime facilitates the generation of low-dimensional slowly varying coherent activity patterns and confers high sensitivity to variations in cortical driving. By re-estimating the model on HD spiking data we discover network parameter modifications are consistent across three very different types of HD mutant mouse models (YAC128, Q175, R6/2). In striking agreement with the known pathophysiology we find feedforward excitatory drive is reduced in HD compared to WT mice, while recurrent inhibition also shows phenotype dependency. We show that these modifications shift the HD MSN network to a sub-optimal regime where higher dimensional incoherent rapidly fluctuating activity predominates. Our results provide insight into a diverse range of experimental findings in HD, including cognitive and motor symptoms, and may suggest new avenues for treatment.

funders had no role in study design, data collection and analysis, decision to publish, or preparation of the manuscript.

**Competing interests:** The authors have declared that no competing interests exist.

## Author summary

Huntington's Disease (HD) is an inherited neurodegenerative disease with devastating symptoms including progressive motor dysfunction and disturbances to normal cognition. The age of disease onset is roughly related to the length of abnormally expanded CAG repeats in the mutant huntingtin gene, but how this produces HD is not well understood. Several transgenic mouse models have been created to investigate the stages of disease progression. HD is found to be primarily associated with pathology of medium spiny neurons (MSNs) in the striatum, the main input stage of the basal ganglia. In wild type (WT) animals MSNs display cell-assembly activation patterns which are known to play a crucial role in striatal cognitive and motor information processing. These activity patterns are lost in HD mice. Here we use computational modeling to probe the role of striatal network dynamics in HD. We fit the parameters of an MSN network model to spiking data from WT mice and three different types of transgenic mice. In agreement with the known pathophysiology, we find cortical feedforward excitation is consistently reduced in all three HD mice. We show how this produces the characteristic dysregulation of MSN activity and explain why it may underlie the motor symptoms of HD.

## Introduction

Huntington's Disease (HD) is an inherited condition caused by an abnormal expansion of CAG trinucleotide repeats in the mutant huntingtin (mHTT) gene (The Huntington's Disease Collaborative Research Group, 1993). HD is a devastating neurodegenerative disease characterized by progressive motor dysfunction and disturbances to normal cognition. In early stages chorea is prominent, while akinesia dominates at later stages [1]. Cognitive symptoms include depression, apathy and anxiety, as well as irritability and agression [2, 3]. The age of disease onset is roughly related to the length of the CAG repeat [4–6]

Several transgenic rodent models of HD have been created [7–9] to investigate the stages of HD progression [10]. Striatal medium spiny neuron (MSN) dysfunction is a prominent finding and may be a pathophysiological cause of HD symptoms. It is well established that MSNs gradually degenerate and eventually die [11] however HD symptoms also occur in the absence of obvious cell loss [12, 13]. Indeed neuronal and synaptic dysfunction precedes cell death by many years in humans [14–16] and occurs before, or even in the absence of, cell death in HD animal models [17–20]. The fact that the Huntingtin protein (Htt) is involved in synaptic function [9, 21] also suggests that HD may primarily be a synaptopathology [22]. Similar alterations of striatal synaptic activity have been demonstrated in multiple different HD mouse models [23]. Progressive changes in spontaneous corticostriatal excitatory synaptic activity [24–27] have been associated with pre- and postsynaptic alterations including reductions in synaptic proteins, loss of dendritic spines, and loss of synapses [1, 9, 24, 28–31]. A progressive disconnection between cortex and striatum seems to be a general characteristic of HD.

In contrast, studies of GABAergic synaptic activity in HD are less conclusive. There are two main types of GABAergic inhibition affecting MSNs in the striatum; feedback via the MSN axon collaterals and feedforward inhibition generated by GABAergic fast spiking interneurons [32]. Although the two types of inhibition have been extensively characterized in WT animals [33–37], little is known about the role of collateral MSN interactions in HD mouse models. GABAergic synaptic activity was found to be modified among a subset of MSNs in HD [4, 38–40] but its source is unknown.

*In vivo* striatal MSNs show strong firing irregularity characterised by very high inter-spike-interval (ISI) coefficients of variation (CV) [41]. Further examination reveals that spike trains are composed of irregularly timed, but comparitively short, bursting episodes of fairly high frequency spiking, usually lasting a few seconds, separated by much longer periods of quiescence or sporadic spiking, lasting tens of seconds [42]. Importantly spiking bursts do not occur in isolation but coherently across multiple MSNs which form cell assemblies *in vivo* [42] and *in vitro* [43]. Because bursting is both coherent across many cells and also occurs on slow behavioural timescales, it is thought to be crucial for striatal cognitive and motor processing. The encoding of movement sequences [44–48], the execution of learned motor programs and sequence learning [49–62] and the representation of time [63–65] all rely on coherent MSN population activity.

Dysregulation of striatal coherent bursting activity is found in various pathological states [66] and in particular is a key component of HD pathophysiology regardless of the severity of HD symptoms, genetic construct, and background strain of the mouse models [42, 67, 68]. WT MSNs do not show intrinsic bursting or subthreshold oscillations when isolated [41, 45, 69–72]. Therefore burst firing in WT mice must be caused by slowly varying fluctuations in input current when the cell is close to spiking threshold. We hypothesized that changes in input current properties may therefore be responsible for the changes in spiking burstiness found in HD.

Previous modeling of the MSN network [73–78] using simple cell models demonstrated that realistic WT spiking characteristics can be reproduced when input current fluctuations are generated by recurrent feedback through the inhibitory collaterals connecting MSN's to each other. Although a rigorous quantitative comparison between model generated and experimental spiking data has never been attempted, a very good qualitative match was obtained, but only when network parameters, such as the IPSP size and MSN to MSN connection probability, were in the vicinity of their physiological measured values [73]. Interestingly, it was found that the MSN network was poised in a critical transition regime [74]. Coherently bursting cell assemblies were shown to reflect an underlying fairly low dimensional dynamical structure which also endowed the MSN network with optimal information processing capabilities, such as the ability to faithfully represent elapsed time, a faculty crucial for the correct sequencing of actions. However outside the physiologically correct parameter regime, coherent bursting cell assemblies were lost, and information processing properties declined.

We wondered if putative synaptic and network alterations underlying the dysregulation of burst firing found in HD could be disambiguated through computational modeling with parameters estimated directly from HD and WT mouse spiking data. In order to make detailed comparisons with data, we studied a large 2500 cell network and used a well-validated specialized single compartment MSN cell model [79, 80]. We employed spiking data from three different transgencic models: R6/2, YAC128 and Q175. The R6/2 mouse model displays a rapidly progressing phenotype, while Q175 and YAC128 progress much more slowly and better represent the stages of the human disease [7, 13, 81–86]. Despite these strong differences in genetic makeup, we find remarkable similarities in spiking dysregulation in the different HD mouse models and in how this dysregulation occurs through synaptic modifications in the estimated MSN network model. In the latter two mouse models, we also investigated age dependency in the spiking data and how this translates into age variation in network model parameters. Again our findings are consistent across the mouse models, despite their different genetic constructs.

The only drug which has been approved for use in HD, tetrabenazine (TBZ), primarily alters corticostriatal synaptic transmission. We hope new insight this modeling provides into

the complex interplay of network and synaptic dysfunctions that take place in HD will stimulate discovery of effective novel drug combinations.

## Results

### Data analysis

Before making detailed quantitative comparison between model generated and experimental data, we first describe salient features of the MSN spiking recordings which vary between WT and HD mice. These quantities are calculated to estimate the fit of the network model to experimental data; however the data analysis also reveals interesting new findings. We investigated spiking data from three different transgenic models: R6/2, YAC128 and Q175. From each model we also investigated data from WT mice of the same background strain, housed in the same environment and of similar age as the HD variants.

R6/2 mice contain a fragment of human mHTT with a large number of CAG repeats (150), and are rapidly progressing. They show motor hyperactivity within 5-7 weeks of birth, akinesia at 8 weeks and die around 12–16 weeks [87–89]. Although some striatal atrophy is found at around 5 weeks [90], MSNs do not die until later [91]. The WT R6/2 mice (R62WT) investigated here were aged between 6 and 9 weeks, while the HD R6/2 mice (R62HD) varied between 8 and 11.5 weeks and were therefore all in the symptomatic stage.

YAC128 mice were created using a yeast artificial chromosome containing the entire full length human mHTT with 128 repeats. The phenotype shows features of the human disease including cognitive and motor dysfunction [81, 92]. Progression is much slower than R6/2 and biphasic with impaired learning at 2 months, motor hyperactivity from 3 months, bradykinesia from 7 months and hypoactivity at 12 months. Modest striatal atrophy is found at 9 months and a small decrease in the quantity of MSNs at 12 months. The YAC128 mice investigated here, both WT (YACWT) and HD (YACHD), vary between 10 and 90 weeks of age and thus extend throughout the symptomatic period.

The recently introduced Q175 knockin mouse [28, 93–95] carries the human mHTT gene in the mouse genomic context [95, 96]. Q175 come in heterozygote and homozygote varieties. Motor, cognitive, and circadian deficits occur in both [94, 97]. In homozygotes, motor symptoms start to develop around 5 months and by 10-12 months they are hypokinetic with mild tremor. By 8 months 10% of striatal volume is lost in heterozygotes and 20% in homozygotes [98]. We analyse data from Q175 WT (Q175WT) mice between the ages of 30 and 60 weeks and Q175 heterozygote (Q175Het) mice between 30 and 90 weeks. We also investigate Q175 homozygote (Q175Hom) data but this dataset is mostly limited to around 30 weeks.

To estimate the model parameters, we only used single unit spiking data. Thirteen of the fifteen quantities used to fit the model to WT and HD data are indicated by the asterisks in Fig 1, which shows spiking characteristics for the seven different mouse types for all ages combined. We find that spiking activity shows consistent changes between WT and HD mice across the three types of HD mutant mice, despite the fact that these mice are very different genetic constructs.

Scaled moments of the ISI distribution are shown in Fig 1(a). In agreement with previous findings [42, 99, 100] we do not find a consistent change in firing rate $r$, mean ISI $\mu$, or ISI rescaled skew, Fig 1(a), between WT and HD mice, while YAC and R6/2 mice do not show any change in $r$ at all. On the other hand, the ISI CV, Fig 1(a), which is the ISI standard deviation $\sigma$ normalized by $\mu$, is very high in all three WT mice and consistently reduced in all three transgenic mice. Spiking much more irregular than Poisson (CV>1) is a well-known salient feature of MSN activity, and its loss is characteristic of HD [42, 67, 68, 101].

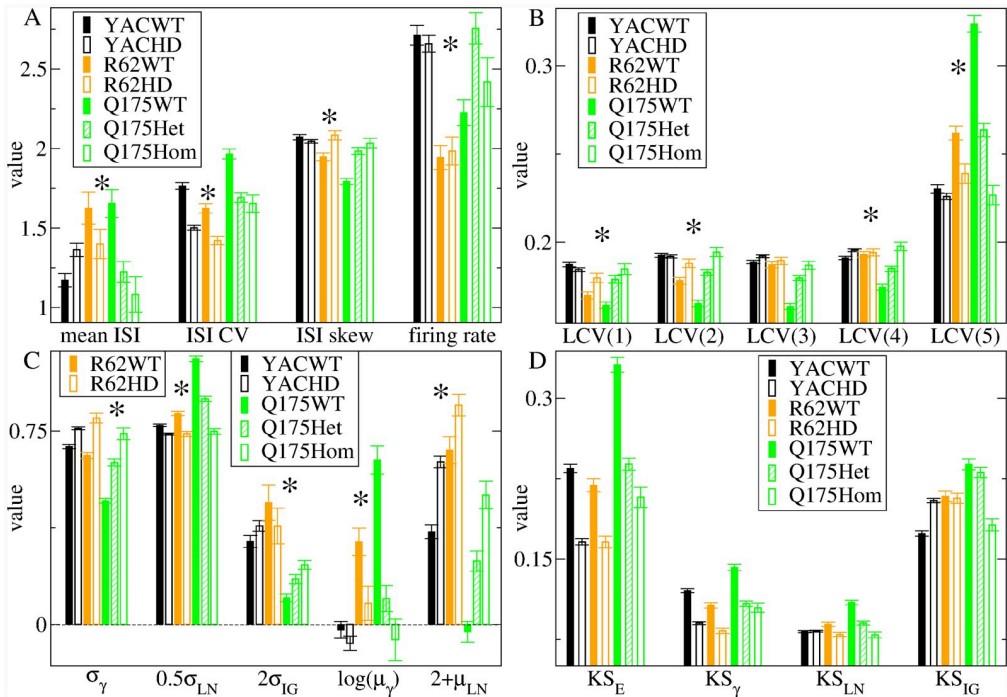

**Fig 1. Experimental data single unit ISI characteristics for the seven different mouse types, (see keys).** (a) mean ISI $\mu$, ISI CV $\sigma/\mu$, ISI rescaled skew S/CV (labelled 'ISI skew'), firing rate $r$, (b) five quintiles of the ISI local CV distribution, LCV(j)$j$ = 1, .., 5, (c) shape parameters for gamma, $\sigma_\gamma$, log normal, $0.5\sigma_{LN}$, and inverse-Gaussian, $2\sigma_{IG}$ distributions, scale parameters for gamma, $\ln(\mu_\gamma)$, and log normal $2 + \mu_{LN}$ distributions (transformations for plotting convenience), (d) KS distance between the data and four maximum likelihood distributions, exponential $KS_E$, gamma $KS_\gamma$, log-normal $KS_{LN}$ and inverse-Gaussian $KS_{IG}$. Results shown are averages of the given quantity calculated from all individual spike trains for each of the seven different types of mice, while bars show SEM (see Methods for a full description of these datasets). The 13 of the 15 quantities which are used to estimate the model parameters (see below) are indicated by the asterisks.

We find that the ISI distribution itself also shows striking and consistent changes between WT mice and their counterpart HD phenotypes. The probability of small ISIs (less than about 1 sec) is reduced in all HD mice, Fig 2(b), 2(d) and 2(f), and this is again progressive in Q175 mouse types, Fig 2(f). The probability of long ISIs may also be increased in Q175WT and R62WT mice compared to HD mice, Fig 2(d) and 2(f). We did not attempt to fit the experimental ISI distribution itself to the model generated distributions but instead compared it with some theoretical distributions often used to generate spiking processes [102], then used these quantities for model parameter estimation. Fig 1(c) shows maximum likelihood (ML) shape ($\sigma_\gamma$, $\sigma_{LN}$, $\sigma_{IG}$) and scale ($\ln(\mu_\gamma)$, $\mu_{LN}$) parameters for three two-parameter distributions: gamma ($\gamma$), lognormal (LN) and inverse-Gaussian (IG). (The IG ML scale parameter is the mean ISI, $\mu$, Fig 1(a)). The $\gamma$ and LN distribution parameters show consistent mouse type dependent changes while the IG parameters do not. Both the reduction in LN shape parameter, $\sigma_{LN}$, and the increase in $\gamma$ shape parameter, $\sigma_\gamma$, in HD compared to WT indicate a change towards a shorter tailed more peaked distribution with fewer very short ISIs and also fewer very long ISIs, in general agreement with direct observations, Fig 2(b), 2(d) and 2(f).

The above measures of the ISI distribution do not depend on the sequential ordering of ISIs. Due to the relevance of burst firing dysregulation in HD, we also use a more detailed measure of burstiness for model estimation, the ISI local CV (LCV) distribution, which depends

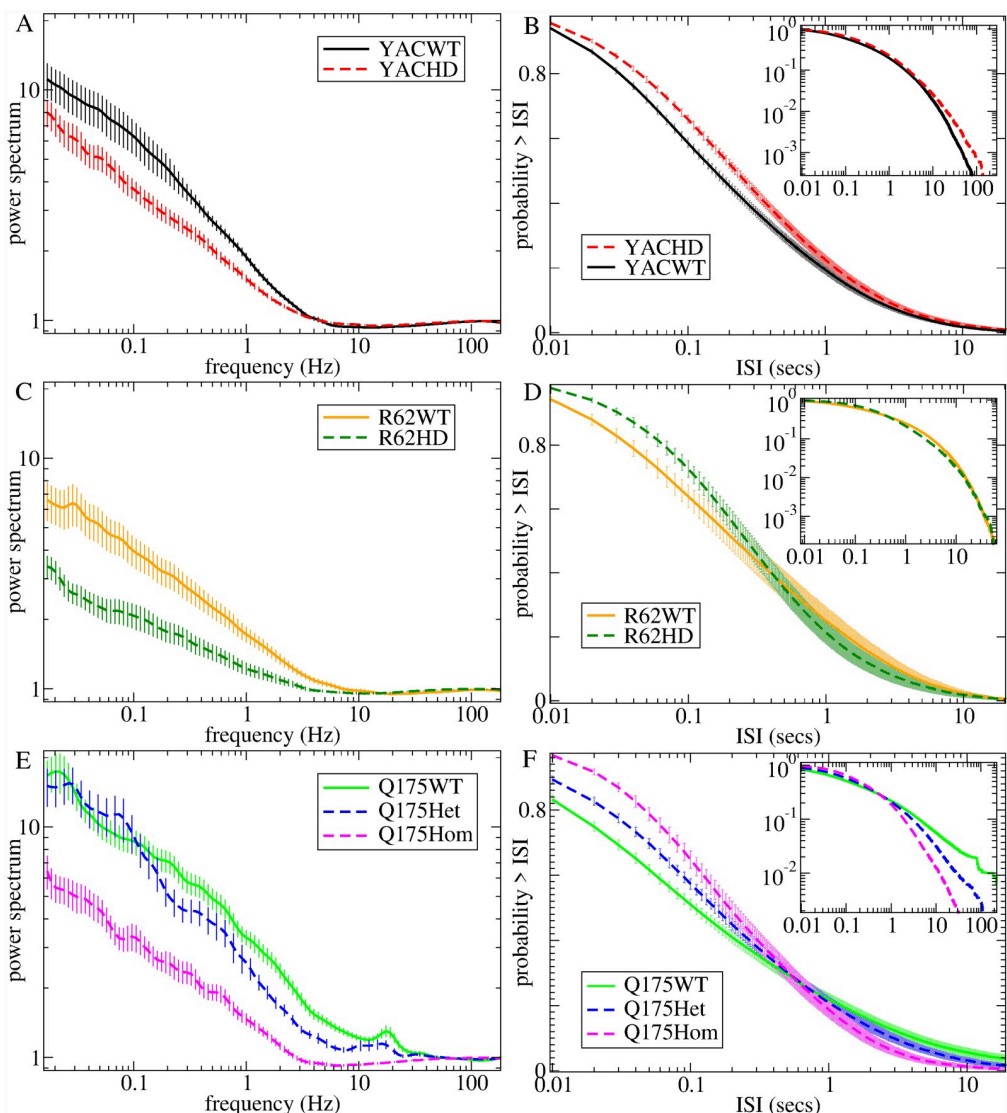

**Fig 2. Experimental data single unit ISI characteristics for the seven different mouse types, (a,b) YACWT, YACHD (c,d) R62WT, R62HD (e,f) Q175WT, Q175Het, Q175Hom (see keys).** (a,c,e) Spike count power spectra, $S(f)$. Power spectra at low frequencies behave like $S(f) \sim f^{-\beta}$ where $\beta$ are YACWT 0.44, YACHD 0.38, R62WT 0.34, R62HD 0.22, Q175WT 0.42, Q175Het 0.5, Q175Hom 0.35. (b,d,f) Cumulative ISI distributions. Main panel: log-linear axes, inset: log-log axes. Results shown are averages across all spike trains for each of the seven different types of mice, while bars show SEM across the spike trains (see Methods).

on the local ISI sequence [103–106]. LCV values are the difference of successive ISI's divided by their sum and bounded between zero and unity (see Methods). The LCV probability distribution has a characteristic shape which is known to depend on recorded cell type and on behavioural and other brain state conditions [103, 107–109] and has been shown to be useful in fitting a spiking network model to data [110]. The LCV distribution divided into five bins of equal size is shown in Fig 1(b). All WT and HD LCV distributions have a profile indicating burst spiking activity as shown by their excess probability (greater than 0.2) of large LCV values, 0.8 < LCV<1, which occur when successive ISIs are very different, compared to low LCV

values, LCV<0.8 (with probability less than 0.2) which occur when successive ISIs are more similar. This bursty profile is consistently attenuated in all three types of HD mice compared to WTs, and the loss is progressive in Q175 types. Finally, we also used two closely related quantities for model estimation: the first two serial lagged ISI autocorrelations (see Methods and below).

Besides these quantities used to fit the model, we also calculated some other partially independent auxiliary quantities to provide a demonstration of how well the estimated best fit model actually predicts other characteristics of the experimental data. The normalized spike train power spectrum, $S(f)$, is shown in Fig 2(a), 2(c) and 2(e) (see Methods). While not completely independent, the power spectrum is not equivalent to the ISI distribution, since the spiking processes we investigate here are not renewal processes, and in fact ISIs show strong long range serial autocorrelations (see below). All power spectra have a characteristic shape dominated by high power at low frequencies, $f$, with a $S(f) \sim f^{-\beta}$ type 'power-law' decay up to about 10 Hz and a minimum somewhere between 10 and 100 Hz (this dip is less pronounced in the Q175WT). The WT power spectra are not identical because the three cohorts of mice differ in strain and age as well as in other respects. Slowly varying activity, with frequency below about 10Hz, is reduced in a very similar way in all three HD mice compared with their WT controls (see $\beta$ values in Fig 2, caption). This reduction also occurs progressively from Q175WT to Q175Het to Q175Hom mice, (though for very low frequencies, <0.1 Hz, the distinction between Q175WT and Q175Het breaks down).

Calculation of the ISI ML quantities described above is always possible and does not suggest the particular theoretical distribution is actually a good fit to the data. We found that the Kolmogorov-Smirnov (KS) distances between the experimental data and the theoretical ML ISI distributions, Fig 1(d), (see Methods), which provide some indication of which distribution is a better fit, also provide a useful, partially independent, measure of how well our estimated best fit model predicts the experimental data, as the KS distances vary strongly with both the experimental data and the model generated data (see below). The LN distribution is clearly the best fit (lowest KS distance) for the three WT mice while the $\gamma$ distribution and LN distribution are almost equally good fits in all three of the HD mutant mice. The IG distribution always has the largest KS distance of the three two parameter distributions. Like the scale and shape parameters themselves, WT/HD dependent changes in $KS_{LN}$ and $KS_{\gamma}$ are consistent across all three mouse models and progressive in Q175, while $KS_{IG}$ variations are not. The KS distance from the one-parameter exponential distribution, $KS_E$, is also shown. This is clearly the worst fit for the WT animals, but comparable to the IG distribution in HD mice.

## Network model

We find various spiking statistical quantities are consistent across different strains of WT mice and also consistent across three different types of HD mutant mice, but many of these quantities also show strong variations between WT and HD mice. We wondered if the particular WT spiking statistics could be quantitatively fit to an MSN network model with physiologically realistic parameter settings and whether modifications to synaptic parameters could account for the changes found in our measures of spiking activity in HD.

In order to make detailed comparisons with data, we employ a well-validated specialized single compartment MSN cell model [79, 80]. This model includes most of the ion-channels thought to be relevant for determining MSN spiking. It captures MSN characteristics [41, 45, 70–72, 111, 112] such as 'up-down' states, whereby a large 20 mV step separates a hyperpolarized resting state from a depolarized state close to firing threshold. It also captures the characteristic long latency to first spike after current injection and provides realistic values for

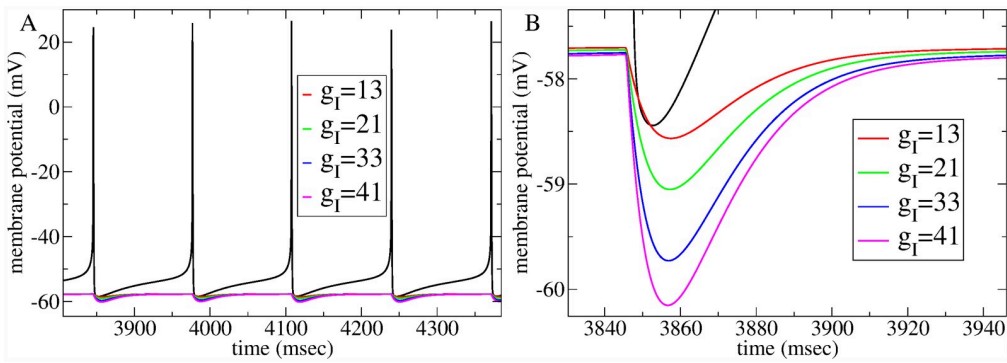

**Fig 3. IPSP size and timescale is in the physiological range at values of $g_I$ found to provide good fits to WT spiking data.** (a,b) Time series of supra-threshold presynaptic spiking cell (black) and slightly sub-threshold postsynaptic cell for different values of $g_I$ (see key), (b) shows detail. At $g_I = 33$, IPSP size is about 1.8 mV and reverts to half maximum in about 20 msec.

rheobase, membrane resitivity, paired-pulse facilitation and other cellular properties. Ion channel parameters are set at the values used in [80] including the modified leak current. We do not use different MSN cell models [113, 114] for dopamine D1 and D2 receptor-expressing types, because our experimental data is not labelled with these MSN types.

In the current work we vary only two network parameters, which are the most important determinants of network activity (see Methods). The first is the level of *net feedforward* driving input current, denoted $g_E$. Each cell receives a fixed random excitatory drive with expectation $g_E$ which, for simplicity, is held constant for the duration of a simulation (see Discussion). This feedforward excitation represents cortical and thalamic driving excitation minus feedforward inhibitory contributions from striatal fast-spiking-interneurons and from any other inhibitory feedforward sources such as the globus pallidus [115].

The second parameter is the strength of *recurrent inhibition* coming from the MSN collateral network, denoted by $g_I$. All inhibitory synapses in the MSN network have a fixed random strength with expectation $g_I$, which determines the size of IPSPs generated by presynaptic spikes in the postsynaptic cell. Inhibitory synapses are modelled in the same way as [80] including a synaptic gating variable dependent on presynaptic membrane potential representing GABA$_A$ neurotransmitter release and decay (see Methods). Fig 3 illustrates spiking in a presynaptic cell and IPSPs in a postsynaptic cell just below firing threshold for several relevant values of $g_I$.

All other network parameters are fixed at their experimentally observed values, such as the cell-cell connection probability (0.2) the network size (2500 cells) and the IPSP rise and decay timescales [80]. Although relevant for striatal information processing [116, 117] we do not include a real spatial dimension, since our experimental single cell spiking data does not include information about relative spatial location between cells, and we find we can well reproduce our single cell statistical quantities without it. Network simulations are entirely deterministic. Since feedforward excitation to each cell is fixed at a constant level, any fluctuations in input current to cells are generated intrinsically by the non-stationary recurrent dynamics of the inhibitory MSN network (see Discussion).

First, in order to appreciate the origin of the spiking characteristics found in the experimental WT data and their dysregulation in HD, we give a rough survey of how network model activity depends on these two parameters. This also demonstrates the wide range of dynamical

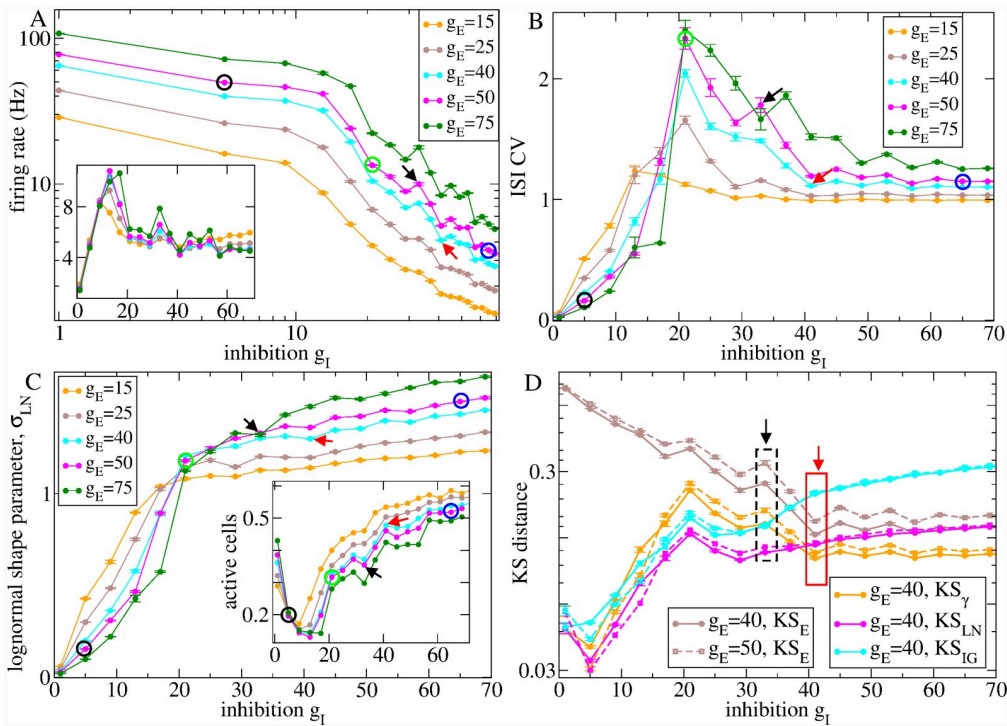

**Fig 4. 2500 cell network model shows transition as parameters are varied.** Several quantities calculated from 200 second network simulations divided into 5, 40 second long, non-overlapping segments. Quantities shown (except c, inset) are calculated from single unit spike trains averaged across all active cells in each 40 second segment. Bars show SEM in these averages across the 5 segments. Results are versus inhibition, $g_I$, at several levels of excitation, $g_E$ (see keys). Simulations at different levels of inhibition, $g_I$, are different realizations of random networks with given connection probability of 0.2. Simulations at any given level of inhibition are identical except for the different levels of excitation, $g_E$. (a) mean firing rate, $r$, (a, inset) rescaled mean firing rate, $r(g_I/g_E)$, (b) mean ISI CV, (c) maximum likelihood lognormal distribution shape parameter, $\sigma_{LN}$, (c inset) number of active cells (which fire at least one spike in the whole 200 second network simulation period), (d) KS distance between the model generated ISI distribution and the estimated maximum likelihood exponential distribution ($KS_E$, brown), $\gamma$ distribution ($KS_\gamma$, orange), lognormal distribution ($KS_{LN}$, pink) and inverse gaussian distribution ($KS_{IG}$, cyan) for two different values of $g_E$ ($g_E = 40$, solid and $g_E = 50$, dashed). (a-c) Black, red, green and blue circles indicate the simulations described in Fig 5. (a-d) Black (WT) and red (HD) arrows (and boxes in (d)) indicate the two individual network simulations which are best-fit for two particular restricted age experimental datasets, denoted 'WT75' and 'HD12'. The ISI statistics for these two simulations and the two corresponding experimental datasets are shown in Figs 8 and 9.

behaviour the MSN network is capable of displaying. Fig 4 shows how spiking activity in network simulations depends on the level of inhibition, $g_I$, and excitation, $g_E$. The firing rate $r$ decreases monotonically as $g_I$ is increased or $g_E$ is reduced, as would be expected, Fig 4(a). However it can clearly be seen that there are two dynamical regimes, which depend on the strength of recurrent inhibition $g_I$. In the *frozen* regime (FR) at low $g_I$, firing rate decreases more slowly than it does in the *active* regime (AR) at higher $g_I$ [74, 77, 78, 118]. The transition between these two regimes is also evident in the number of active cells, Fig 4(c), which are those which fire at least one spike during the 200 second simulations. This has a minimum as $g_I$ increases for all levels of $g_E$ [74, 77, 78]. The presence of two regimes is also visible in the ISI CV which is non-monotonic, peaking at an intermediate $g_I$, Fig 4(b), for all $g_E$.

This transition in network dynamics [74, 78, 102, 119–122] has recently been understood using the powerful tools of dynamical mean-field theory (DMFT) [123–125]. When IPSPs decay fairly slowly on the timescale of spiking, as they do here, individual spiking events are

averaged out and network dynamical behaviour is determined by the associated dynamics of the mean input current to cells. In the FR, mean input currents are stationary and unvarying [123, 124]. As inhibition, $g_I$, is increased, at some point this stable fixed point state becomes unstable and a non-stationary state, the AR, with chaotically varying input currents appears [74, 78, 123, 124]. Just above the transition, input current fluctuations are 'critical' and while small in magnitude their autocorrelation decays exceedingly slowly [123, 124]. Further into the AR, fluctuations increase in size dramatically, but also decay more rapidly.

The DMFT approximation to the spiking network dynamics is relevant even when synaptic decay timescales are as small as 20 ms and cells have biologically reasonable firing rates [123]. However the presence of spiking modifies the predictions of DMFT somewhat. Even in the FR, small fluctuations arise from spiking events which decorate the fixed point state, and these fluctuations can be non-linearly amplified as $g_I$ is increased and the transition to the AR is approached. In the critical regime, spiking noise prevents full divergence of the autocorrelation timescale. Thus spiking smoothes the transition [124]. The DMFT predicts that due to the dynamical balance between recurrent inhibition and feedforward excitation, firing rates should be proportional to $g_E/g_I$ [123, 124]. Network simulation firing rates rescaled by this quantity are shown in Fig 4(a, inset). While there are substantial deviations, the approximate independence of the rescaled firing rate from $g_E$ and $g_I$ demonstrates the relevance of the DMFT theory, even in this spiking network model of fairly physiologically detailed MSN cells.

Fig 5 illustrates how network spiking characteristics are determined by DMFT for three simulations at different levels of recurrent inhibition, $g_I$, and fixed excitation, $g_E = 50$ (indicated by coloured circles in Fig 4(a)–4(c)). Fig 5(a) shows the mean ISI versus the ISI CV for all active cells in these simulations. The black points illustrate a simulation in the FR with low $g_I = 5$. The FR is a winners-take-all (WTA) like state. The winners are the cells most strongly excited (by feedforward excitation minus feedback inhibition). They remain permanently above firing threshold and spike highly regularly suppressing the losers permanently below firing threshold. Thus almost all active cells have small mean ISI $\mu$ and very low CV $\sim 0$, although there are also a few cells whose mean input current is just below firing threshold which spike occassionally with very low rates (large $\mu$) in a Poisson-like way, CV $\sim 1$. The corresponding ISI distribution averaged across active cells, Fig 5(b, black), contains mostly small ISIs from the regularly firing cells with a very shallow shoulder contributed by active cells. Power spectra averaged across cells, Fig 5(d, black), are dominated by the regularly firing cells with a peak around their maximal firing rate, 80 Hz, and absence of lower frequency fluctuations [102, 119, 120]. For comparison with experimental data, we also calculate ISI serial autocorrelations (see Methods). Fig 5(c, black) shows this quantity averaged across active cells for the low $g_I = 5$ simulation in the FR. Autocorrelations are short term negative, small and positive thereafter and decay to zero fairly rapidly. In this purely inhibitory network, all cells have a fixed minimum ISI determined by their fixed levels of driving excitation $g_E$. This absolute refractory period results in the negative short term autocorrelations in the FR and also produces a decrease in CV with increasing driving excitation $g_E$ at $g_I = 5$ in the FR, Fig 4(b) [102]. Consistent with this, the lognormal distribution shape parameter, $\sigma_{LN}$, is below unity in the FR, Fig 4 (c), suggesting that ISI distributions are sharply peaked at non-zero ISI and absent small and long ISIs.

As the level of inhibition, $g_I$, is increased in the FR, more and more cells get suppressed below threshold, and the quantity of active cells at first decreases, Fig 4(c, inset). However as $g_I$ increases further, at some point the DMFT fixed point loses stability and input currents start to wander in a non-stationary way. The dramatically larger fluctuations cause the quantity of active cells to rapidly increase again, Fig 4(c, inset). In the critical AR above the transition at $g_I = 21$, Fig 5(a-d, green), bursty cells with very high CV appear, Fig 5(a, green). Slow fluctuations

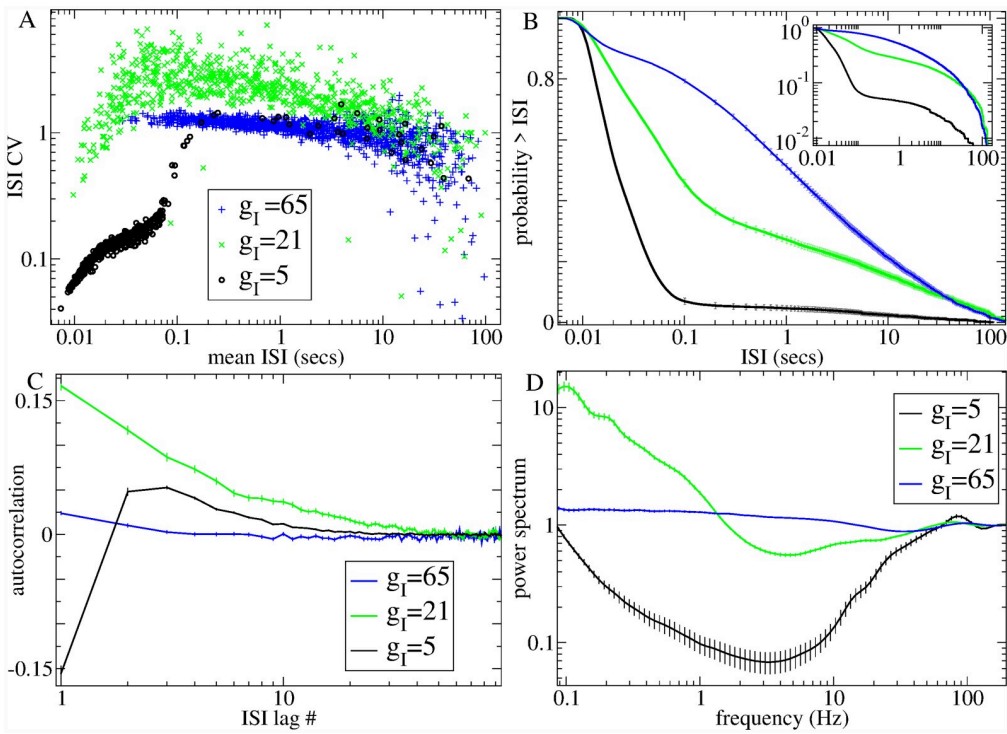

**Fig 5. Model generated single unit ISI characteristics for three illustrative simulations with different levels of inhibition, $g_I$ (see keys) at a fixed level of excitation, $g_E$ = 50, indicated by the coloured circles in Fig 4(a)–4(c).** (a) Mean ISI, $\mu$, versus ISI CV for all individual active cells in each simulation, (b) Cumulative ISI distribution, (c) ISI serial lagged autocorrelation, (d) Spike power spectra, $S(f)$. (b-d) Results are averages across the given quantity calculated from all individual single cell spike trains with more than 10 spikes in each 200 second simulation and bars show SEM across these observations (see Methods).

in inhibitory input current cause cells to irregularly switch between regularly spiking and quiescent states. The ISI distribution, Fig 5(b, green), contains a broad exponential shoulder reflecting the long interburst intervals and a peak at small ISI from the intraburst ISIs. ISI serial autocorrelation, Fig 5(c, green), is positive for all lags and decays very slowly due to the long bursts of spikes while spectral power, $S(f)$, at low frequencies increases dramatically, Fig 5(d, green) and displays a power-law form, $S(f) \sim f^{-\beta}$ reflecting the very long timescales which appear. In the AR, the lognormal distribution shape parameter, $\sigma_{LN}$, Fig 4(c), shows a transition to a value exceeding unity, indicating the presence of arbitrarily small ISIs and a long tail. In contrast to the FR, now increasing the driving excitation $g_E$ increases the CV, Fig 4(b). In the AR, input current fluctuations vary on a much longer timescale than the spiking itself. Increasing excitation results in an increase in the amount of spikes within a burst without a large effect on the interburst intervals, which increases the CV. In the critical regime, the effect of a small change in $g_E$ on CV can be substantial.

Finally as recurrent inhibition increases further to $g_I$ = 65, Fig 5(a-d, blue), mean-field input current fluctuations increase in magnitude but their autocorrelation decays more rapidly [123, 124]. All cells fluctuate rapidly between activity and quiescence. Spiking therefore approaches Poisson for almost all cells as demonstrated by the fact that CV values are close to unity, Figs 4(b) and 5(a, blue), the ISI distribution Fig 5(b, blue), is almost exponential, ISI serial autocorrelation, Fig 5(c, blue), is absent and the power spectrum, $S(f)$, Fig 5(d, blue), flattens out.

We also investigated how the KS distances between model generated ISI distributions and the theoretical ML exponential, gamma ($\gamma$), log-normal (LN) and inverse-Gaussian (IG) distributions, as described above, vary with the model parameters. Fig 4(d) shows the KS distances versus model recurrent inhibition, $g_I$, for two values of driving excitation, $g_E = 40$ and $g_E = 50$. The KS distances demonstrate interesting crossovers close to the transition from FR to AR. At very low $g_I$ in the FR for any given $g_E$, the LN distribution (pink) is definitely the best fit (lowest KS distance) while the $\gamma$ (orange) and IG (cyan) distributions are similar, $\text{KS}_{LN} < \text{KS}_\gamma \approx \text{KS}_{IG}$. Increasing $g_I$, in particular for higher $g_E$, we find a regime where $\text{KS}_{LN} < \text{KS}_{IG} < \text{KS}_\gamma$, while as $g_I$ is increased further a regime with $\text{KS}_{LN} < \text{KS}_\gamma < \text{KS}_{IG}$ appears. Finally in the AR the $\gamma$ distribution becomes definitely the best fit, $\text{KS}_\gamma < \text{KS}_{LN} < \text{KS}_{IG}$.

## Model estimation

We find various differences between WT and HD data. Many of these differences are consistent across phenotypes. The above observations of network model transitions suggested to us that differences between WT and HD data might be explained by changes in network model parameters, $g_E$ and $g_I$. However the non-monotonic dependence of ISI statistical quantities on model parameters means multivariate measures must be used for estimation.

Here we take a very direct approach. We calculate several ($N$) statistical measures, termed features, from each of the single spike trains in a particular dataset, for example the YACHD dataset. This results in a multivariate distribution where each $N$–dimensional observation corresponds to one spike train. We also calculate the exact same set of features from each of the spike trains generated from a 2500 cell network model simulation with some given parameter values, $g_E$ and $g_I$. This results in another $N$–dimensional multivariate distribution characterising the given model. We then use the Kullback-Leibler (KL) distance between these two distributions as a measure of how far the given model simulation is from the particular experimental dataset (see Methods). The fifteen features used in this calculation are the thirteen indicated by asterisks in Fig 1 and the first two lagged ISI serial autocorrelations (see Methods).

Heat maps of the natural logarithm of the KL distance, log KL, for the seven experimental datasets with all animals of a given type of all ages combined, as well as two restricted age datasets (see below), are shown in Fig 6. Here each large coloured matrix corresponds to a particular experimental dataset and each small square in each matrix shows the KL distance from the experimental dataset to a network model simulation parameterised by $g_I$ and $g_E$. Note that the $g_E$ (vertical) axis is not regularly spaced. Similarities and differences are evident in these heat maps, reflecting the similarities and differences we found in the original data. The first column, Fig 6(a), 6(d) and 6(g), shows the WT datasets, (a) Q175WT, (d) YACWT, (g) R62WT while the second column Fig 6(b), 6(e) and 6(h), shows the HD datasets, (b) Q175Het, (e) YACHD, (h) R62HD and the top of the third column Fig 6(c) shows Q175Hom. The three WT heatmaps Fig 6(a), 6(d) and 6(g) are very similar to each other with a minimum (best fit of the KL distance) around $g_I = 25 \sim 41$ and $g_E = 50 \sim 75$. These $g_I$ values place the WT datasets close to the transition, but definitely in the AR in Fig 4, as previously suggested [73, 74, 77, 78] based on qualitative observations. However, models only provide good fits to WT datasets at higher levels of $g_E$, even though the FR-AR transition is found at all $g_E$ levels.

HD datasets Fig 6(b), 6(e), 6(h) and 6(c) on the other hand display a different but consistent profile. In all cases the best fit area of minimum KL moves to lower values of $g_E$ and this change is progressive in the Q175 mouse types, Fig 6(a), 6(b) and 6(c). The area of best fit inhibition also becomes more diffuse and spread over a wider range $g_I = 33 \sim 57$. That consistent changes in these heat maps are found across different strains of wild type mice and different

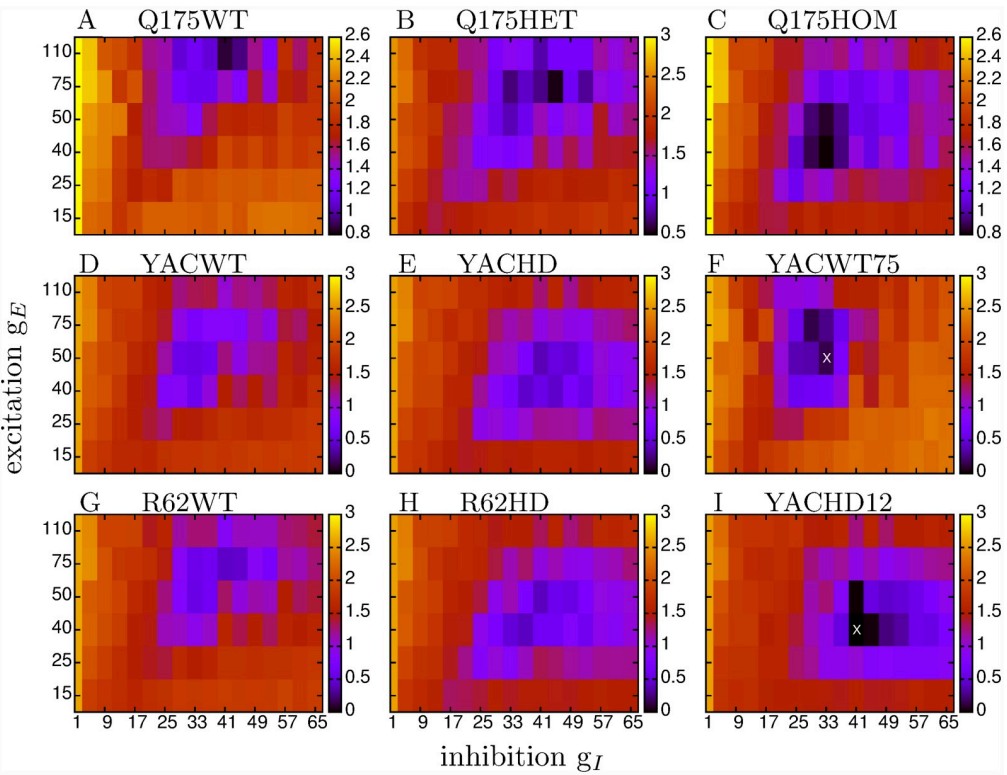

**Fig 6. Heat maps showing goodness of fit measure, log(KL), (the lower the better), for multiple network simulations compared to different experimental datasets.** (a) Q175WT, (b) Q175Het, (c) Q175Hom, (d) YACWT, (e) YACHD, (g) R62WT, (h) R62HD, (f) restricted age YACWT dataset denoted 'WT75', (i) restricted age YACHD dataset denoted 'HD12'. In each panel the x-axis shows network model inhibition, $g_I$, and the y-axis network model excitation, $g_E$. Note $g_E$ is not equally spaced. Each small square, corresponding to a particular $g_I$, $g_E$ pair, in each panel shows the KL distance, log(KL), for that network simulation from the particular experimental dataset. The bars on each panel show the log(KL) colourscale. (f,i) These two datasets are indicated by the pink arrows in Fig 11(a) and 11 (b) and their spiking characteristics are shown in Figs 8, 9(a), 9(c) and 9(e). The crosses indicate the best fit model simulations whose spiking characteristics are shown in Figs 8, 9(b), 9(d) and 9(f).

types of HD mutants is very striking. Evidently the dysregulation seen in HD is caused by a shift away from the transition regime further into the AR.

To make this more explicit we calculated weighted average values, denoted by $g_I^*$ and $g_E^*$, over the $g_I$ and $g_E$ parameters using a decreasing function of the KL distance as the weighting. Roughly speaking, this provides the location of the minimum in the KL heat maps, Fig 6, which is the network model simulation best fit for the dataset (see Methods, where it is also demonstrated that this method works well to determine $g_E^*$ and $g_I^*$ in this network model). Fig 7 shows the weighted average $g_E^*$ and $g_I^*$ obtained from the heatmaps, Fig 6(a)–6(c), 6(d), 6 (e), 6(g) and 6(h). We find a consistent, strong and significant decrease in feedforward excitation, $g_E^*$, across all three HD mouse types, which is progressive in Q175, from WT to heterozygous to homozygous, and an increase in inhibition, $g_I^*$, in YAC mice. Interestingly, we find $g_E^*$ is very similar across the three HD mouse types (YACHD, R62HD, Q175Hom). but somewhat different across the three WT mouse types.

## Comparison of experimental and model generated data

How well do spike trains generated by the best fit network models resemble the experimental data? The combined mouse type datasets described above contain spike train recordings made

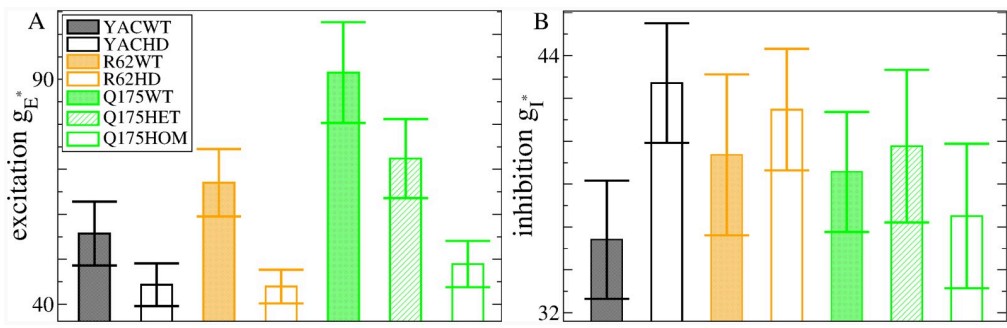

**Fig 7. Model estimated best fit (a) excitation $g_E^*$ and (b) inhibition $g_I^*$ for the seven different mouse type datasets YACWT, YACHD, R62WT, R62HD, and Q175WT, Q175Het and Q175Hom (see key on (a)).** Bars show weighted standard errors in estimated quantities (see Methods).

over multiple weeks and include a large amount of variability due to the progression of disease as well as other age related changes. To make more accurate fits to the experimental data, we divided our spike trains into distinct non-overlapping datasets in restricted age ranges.

Fig 6(f) and 6(i) show, as an illustration, two examples of KL heat maps for restricted age datasets. Fig 6(f) shows the KL heat map for a YACWT dataset made up of all spike trains within the three week interval between 75 and 78 weeks, denoted WT75, while Fig 6(i) shows the KL heat map for a YACHD dataset including all spike trains with ages between 12 and 15 weeks, denoted HD12. These experimental datasets were chosen as illustrations because they have multiple ISI statistics that differ between them (see below). As expected, these restricted age heat maps, Fig 6(f) and 6(i), appear to have stronger more focused peak values than the heat maps constructed from the whole YAC datasets Fig 6(d) and 6(e), since the whole YAC datasets contain cells recorded from 10 to 90 weeks.

The crosses in Fig 6(f) and 6(i) show the best fit network models which have the minimum KL values for these datasets. In these particular examples, the best fit WT model has lower recurrent inhibition, $g_I^* = 33$, and higher feedforward excitation, $g_E^* = 50$, than the best fit HD model, $g_I^* = 41$ and $g_E^* = 40$, as is the case for the combined datasets comprising all YACWT and YACHD observations, Fig 7. These best fit network models are indicated by the arrows in Fig 4(a)–4(c). To demonstrate the quality of these fits, we now compare model generated and experimental data sets.

Fig 8 shows the same statistical quantities as described above in Fig 1, now for the WT75 and HD12 experimental datasets (solid bars) compared to their best fit WT and HD model generated datasets (empty bars). Various strong differences between the experimental HD and experimental WT data (red and black solid bars) can be seen. For example the ISI CV, Fig 8(a), is much lower in HD than WT animals and the HD LCV(j) profile, Fig 8(b), is strictly increasing with $j$ while the WT profile has a minimum. Many of these differences reflect those found in the larger datasets made up of all YAC recordings, Fig 1(black solid, black empty). Again, the LN distribution is clearly the best fit ISI distribution (lowest KS distance) for WT animals, Figs 8(d) and 1(d), whereas the gamma distribution is slightly better in HD animals.

Evidently the model fits are mostly excellent. In many cases model values (empty bars) are very close to experimental values (solid bars), for example both WT and HD ISI CV, Fig 8(a). In other cases, the relative form of the experimental data is captured by the best fit model. For example the HD and WT LCV(j) experimental profiles, Fig 8(b), including the minimum in the WT, are well represented in the best fit models. The higher IG shape, $\sigma_{IG}$, lower gamma

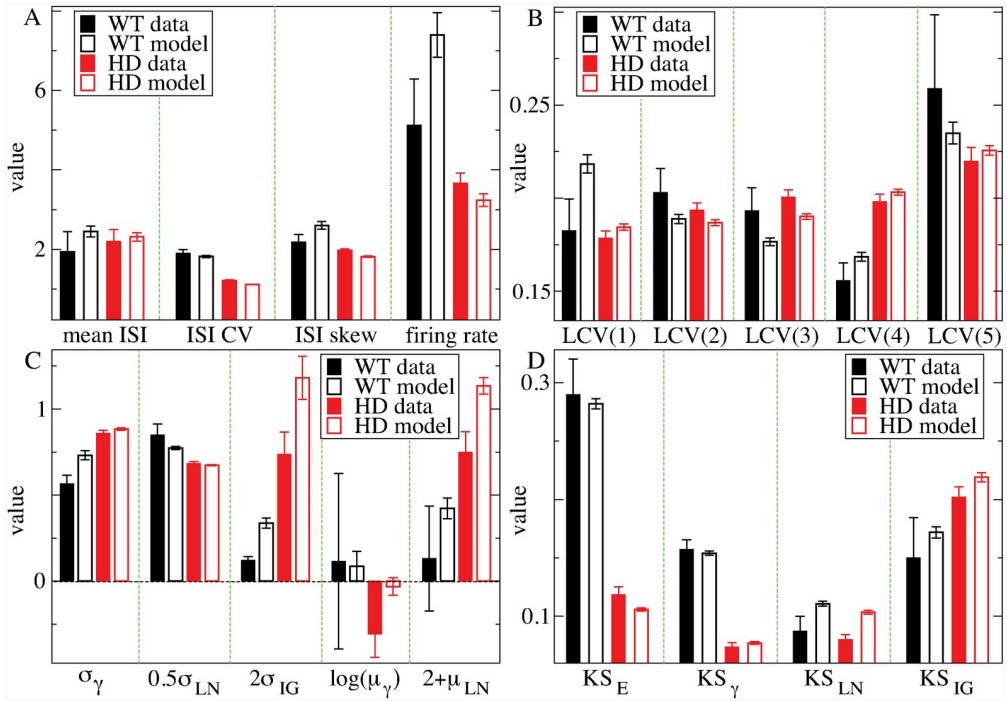

**Fig 8. Comparison of ISI statistical quantities for the YACWT restricted age dataset denoted, 'WT75', (solid black), and the YACHD restricted age dataset denoted 'HD12' (solid red), whose log KL heat maps are shown in Fig 6(f) and 6(i) and whose ISI characteristics are shown in Fig 9(a), 9(c) and 9(e), and the same summary statistics from the best fit network models (WT empty black, HD empty red) indicated by crosses in Fig 6(f) and 6 (i) whose ISI characteristics are shown in Fig 9(b), 9(d) and 9(f).** ISI statistical quantities are the same as those described in Fig 1. (a) mean ISI $\mu$, ISI CV $\sigma/\mu$, ISI rescaled skew S/CV (labelled 'ISI skew'), firing rate, $r$, (b) five quintiles of the ISI local CV distribution, LCV(j)$j = 1, .., 5$, (c) shape parameters for gamma, $\sigma_\gamma$, log normal, $0.5\sigma_{LN}$, and inverse-Gaussian, $2\sigma_{IG}$, distributions, scale parameters for gamma, $\ln(\mu_\gamma)$, and log normal, $2 + \mu_{LN}$, distributions (transformations for plotting convenience), (d) KS distance between the data and four maximum likelihood distributions, exponential, $KS_E$, gamma, $KS_\gamma$, log-normal, $KS_{LN}$, and inverse-Gaussian, $KS_{IG}$. Results shown are averages across all individual spike trains with more than 10 spikes in each of the four datasets (two experimental and two model) and bars show the SEM in these averages (see Methods).

scale, $\mu_\gamma$ and higher LN scale, $\mu_{LN}$ parameter relationships in HD compared to WT experimental data, Fig 8(c), are all correctly represented by the models. Most interestingly, ISI characteristics which are not themselves features used in the model estimation procedure (asterisks in Fig 1), like the KS distance measures, Fig 8(d), are also very close to experimentally observed values. This suggests the model is in fact an accurate description of the data.

Fig 9(a), 9(c) and 9(e) shows the power-spectra, ISI distribution, and ISI serial autocorrelation for the same restricted age experimental YAC datasets, WT75 (black) and HD12 (red). These groups show the same characteristic loss of low frequency power (<1 Hz) and reduction in probability of short ISIs (<0.1 sec) for HD (red) compared to WT (black) found consistently in the whole group experimental data, Fig 2. The WT ISI distribution, Fig 9(c), shows a shoulder which is absent in the HD distribution, and there is a dip in the power-spectra around 10 Hz which is missing in the HD data, Fig 9(a). Both WT and HD data show positive ISI serial autocorrelations, Fig 9(e), extending over several tens of lags, which however are strongly reduced in magnitude in HD compared to WT.

The corresponding results from the best fit network models described above are shown in Fig 9(b), 9(d) and 9(f). The agreement is again excellent, quantitatively as well as in general

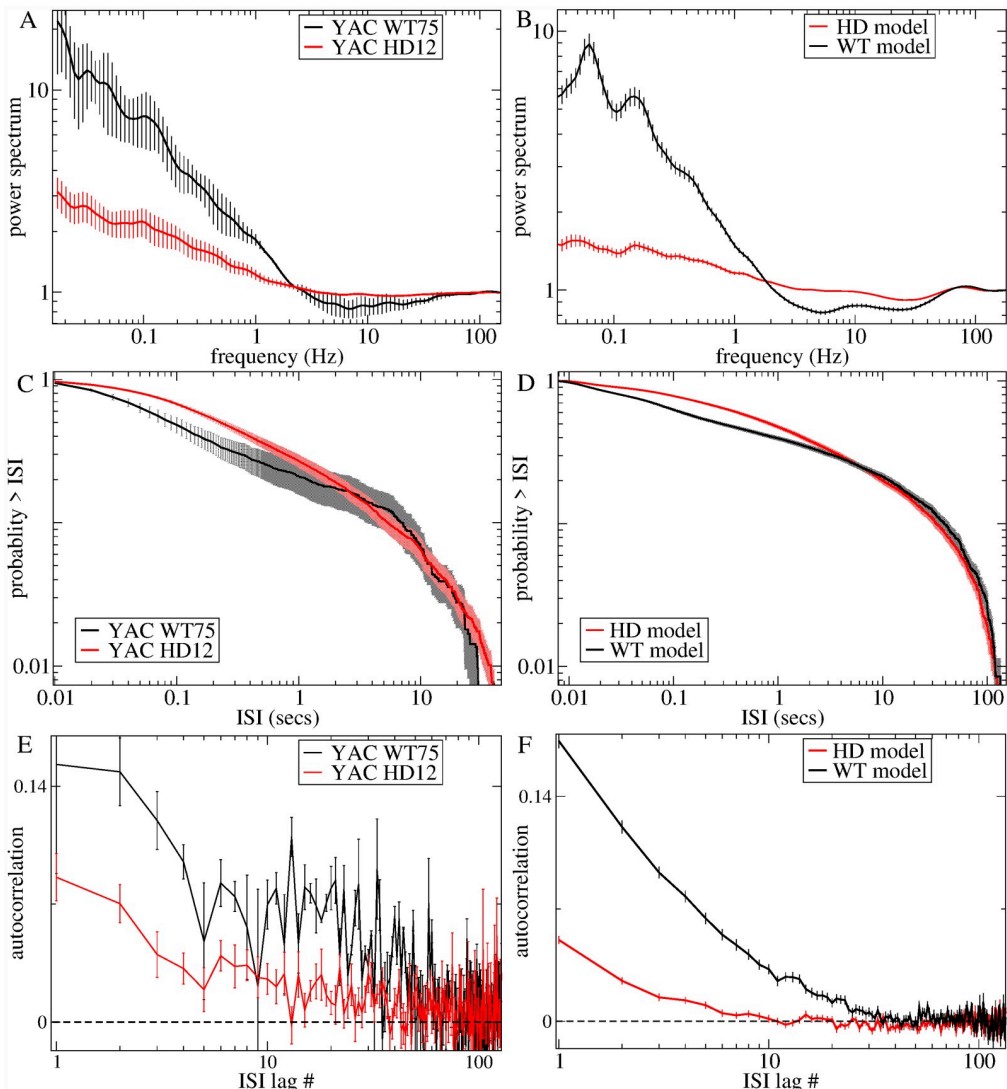

**Fig 9. Comparison of experimental data and best fit model generated data.** (a,b) Spike count power-spectra. (c,d) Cumulative ISI distribution. (e,f) Lagged ISI serial autocorrelation function (a,c,e) Experimental data for two age restricted datasets, the YACWT group, denoted 'WT75' (black) and the YACHD group, denoted 'HD12' (red). Results shown are averages across all the spike trains in each group and bars show SEM across these spike trains (see Methods). (b,d,f) Network simulation generated data with parameter settings WT: $g_E$ = 50, $g_I$ = 33 (black), HD: $g_E$ = 40, $g_I$ = 41 (red) which are best fit for the experimental group datasets, WT75, and HD12, shown in (a,c,e). Results are averages across all active cells in each network simulation and bars show the SEM (see Methods).

form. The ISI distribution shows the reduction in probability of short ISIs (<0.1 sec) for HD (red) compared to WT (black) and absence of the shoulder in HD, Fig 9(d). Power-spectra show the loss of low frequency power in HD compared to WT, and the 10 Hz dip is present in WT but not in HD, Fig 9(b). ISI serial autocorrelations, Fig 9(f), for these two models are positive and decay over tens of lags, as in the experimental data, while the magnitude is strongly reduced in HD compared to WT, as in the experimental data. It is remarkable that we are able to predict the full temporal structure of these ISI characteristics when the model is estimated only on quantities that far from fully constrain them. These are non-trivial findings given the

large range of possible profiles these quantities can take up in this network model, as demonstrated in Fig 5, and again suggest that the model is correct.

These two WT and HD best fit models are quite close in parameter space, only separated by $\Delta g_E = 10$ and $\Delta g_I = 8$. The small difference between IPSPs at $g_I = 33$ (WT) and $g_I = 41$ (HD) is illustrated in Fig 3(b). How does such a striking change in spiking dynamics occur? Fig 4 illustrates that a small change in parameters can have a large effect when parameters are modified in the vicinity of the network transition. The arrows, Fig 4(a)–4(c), and rectangles, Fig 4(d), indicate the best fit WT (black) and HD (red) models.

Close to the transition, ISI CV falls dramatically from a very bursty WT value of about 1.7 to a minimal Poisson-like HD value about 1.1, as $g_E$ is reduced and $g_I$ increased, Fig 4(b). On the other hand, the drop in firing rate, Fig 4(a), from about 7Hz for the WT model to about 3Hz for the HD model is rather modest considering the large range possible within the $g_E$ and $g_I$ models explored. The ISI distribution KS measures, Fig 4(d), reveal the effect of the transition particularly clearly. The KS distance measures for the WT best fit model ($g_E = 50$, $g_I = 33$), Fig 4(d, dashed lines within black dashed rectangle), show that the LN KS distance is a clear minimum, indicating that the WT model is in the transition regime. Increasing $g_I$ and decreasing $g_E$ just a small amount to the HD best fit model parameter values ($g_E = 40$, $g_I = 41$), Fig 4(d, solid lines with solid red rectangle), causes a transition to a state where the $\gamma$ distribution is the minimum.

## Age dependency

YAC and Q175 HD mice display a slowly progressing phenotype with behavioural changes across the age range from 10 to 100 weeks [13, 81, 86, 92, 94, 97]. We next wondered if we could find a progressive variation in statistical quantities with age, which might possibly be associated with disease progression. Fig 10 shows how some of the ISI statistical quantities examined above, Fig 1, depend on mouse age for YACWT, YACHD, Q175WT and Q175Het. Spike trains from YAC mice were divided into four groups based on 23 week intervals with ages in the range $23i \sim 23(i + 1)$ where $i \in \{0, 1, 2, 3\}$. Since Q175 mice were recorded over a smaller span of ages we divided our Q175 spike train dataset into 10 week age intervals with ages in the range $10i \sim 10(i + 1)$ (see Methods). This resulted in four Q175WT datasets, where $i \in \{2, 3, 4, 5\}$ and six Q175Het datasets, where $i \in \{2, .., 7\}$.

We find surprisingly strong similarities in age progression between the two HD mouse types which contrast with the WT data. Both YACWT, Fig 10(a, c, black circles) and Q175WT Fig 10(b, d, green circles), ISI statistical measures show some quite large fluctuations with age but do not show clear trends. On the contrary both the YACHD and Q175Het mouse types, Fig 10(red and blue squares), show several consistent and significant trends. Mean ISI, $\mu$, decreases while ISI CV and skew increase, Fig 10(a red squares, b blue squares). In both YACWT and Q175WT mice consistently in every age group (8 age group datasets in total) KS distances are maintained in the network model transition regime, $KS_{LN} < KS_\gamma < KS_{IG} < KS_E$, Fig 10(c black circles, d green circles). However this is only true for the oldest group of YACHD mice Fig 10(c, red squares), and the oldest two groups of Q175Het mice, Fig 10(d, blue squares). In younger HD mice the $\gamma$ distribution is either the best fit or as good a fit as the LN distribution. The change occurs through a general increase of $KS_\gamma$ and decrease of $KS_{LN}$ and $KS_{IG}$ with age in both YACHD and Q175Het mice. These experimental data indicate that, intriguingly, rather than progressively moving away from the WT phenotype with age, HD MSN ISI characteristics seem to be pathological when younger but normalize with age. This may be a compensatory mechanism (see Discussion).

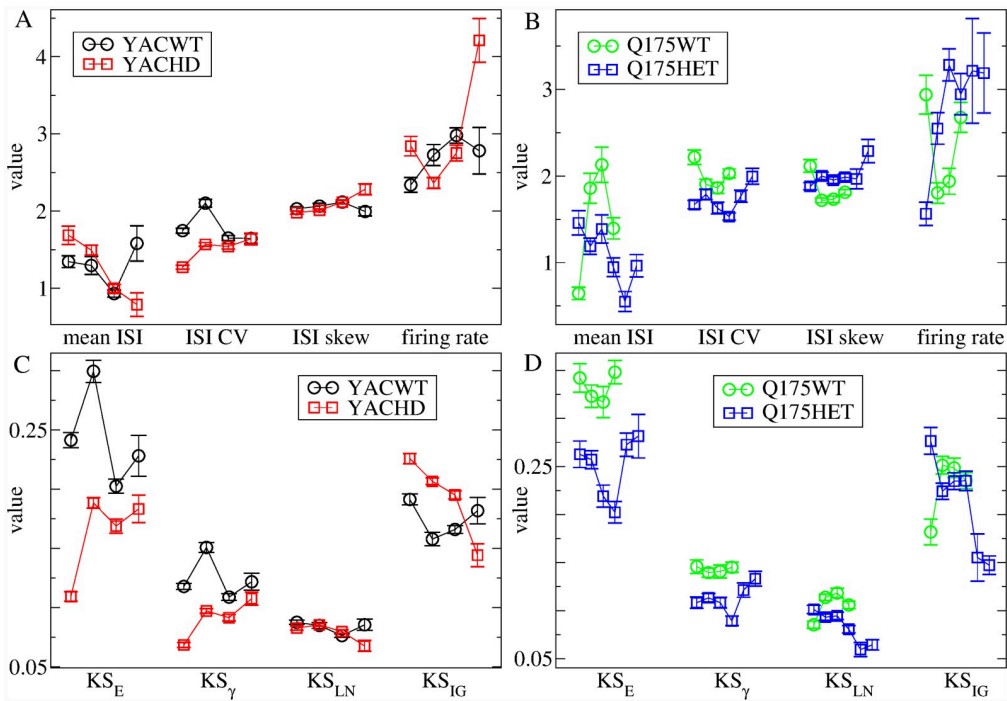

**Fig 10. ISI statistical quantities calculated from experimental spike trains divided into distinct age intervals.** (a,c) YACWT (black circles) and YACHD (red squares) spike trains are divided into four 23 week intervals, 0-23, 23-46, 46-69, and over 69 weeks. (b,d) Q175WT (green circles), spike trains are divided into four 10 week intervals, 20-30, 30-40, 40-50 and 50-60 weeks. Q175Het (blue squares), spike trains are divided into six 10 week intervals, 20-30, 30-40, 40-50, 50-60, 60-70 and 70-80 weeks (see Methods). (a,b) mean ISI $\mu$, ISI CV $\sigma/\mu$, ISI rescaled skew S/CV (labelled 'ISI skew'), firing rate, $r$. (c,d) KS distance between the data and four maximum likelihood distributions, exponential, $KS_E$, gamma, $KS_\gamma$, log-normal, $KS_{LN}$, and inverse-Gaussian, $KS_{IG}$. Values shown are averages of the given quantity across all the spike trains in the particular dataset and bars show SEM (see Methods).

We next asked how these trends in the data translated into trends in network model parameters. For this, we divided the YAC and Q175 spike trains into smaller non-overlapping three week interval datasets with ages in the range $3i \sim 3(i + 1)$, $i = \{0, 1, ..\}$ weeks. For each of these restricted age group datasets we calculated the weighted average values, $g_E^*$ and $g_I^*$, indicating the best fit network model parameters as described above. These values, for each of the YAC and Q175 datasets, are shown in Fig 11(a)–11(d) versus the mean age of the spike trains in the dataset.

We find best fit feedforward excitation, $g_E^*$, is larger in almost all YACWT datasets, Fig 11(a, black circles) than the corresponding YACHD datasets of the same age, Fig 11(a, red crosses). The same is true for Q175WT, Fig 11(c, green circles), versus Q175Het, Fig 11(c, blue crosses). Our Q175Hom dataset, although very limited, showed further reduced excitation at all ages, Fig 11(c, pink squares). This recapitulates what was found for the combined datasets, Fig 7(a). Feedforward excitation, $g_E^*$, shows large fluctuations across datasets in all mouse types, in particular for YACWT, Fig 11(a, black circles). Despite this, we performed linear regression fits of the datasets versus age (see Methods) and, as shown by the slopes in the figure insets and p-values in the caption, we found significant trends towards increasing $g_E^*$ with age in both YACHD, Fig 11(a, inset, red cross), and Q175Het datasets, Fig 11(c, inset, blue cross). YACWT feedforward excitation, $g_E^*$, also showed a significant trend to decrease with age, Fig 11(a, inset, black circle).

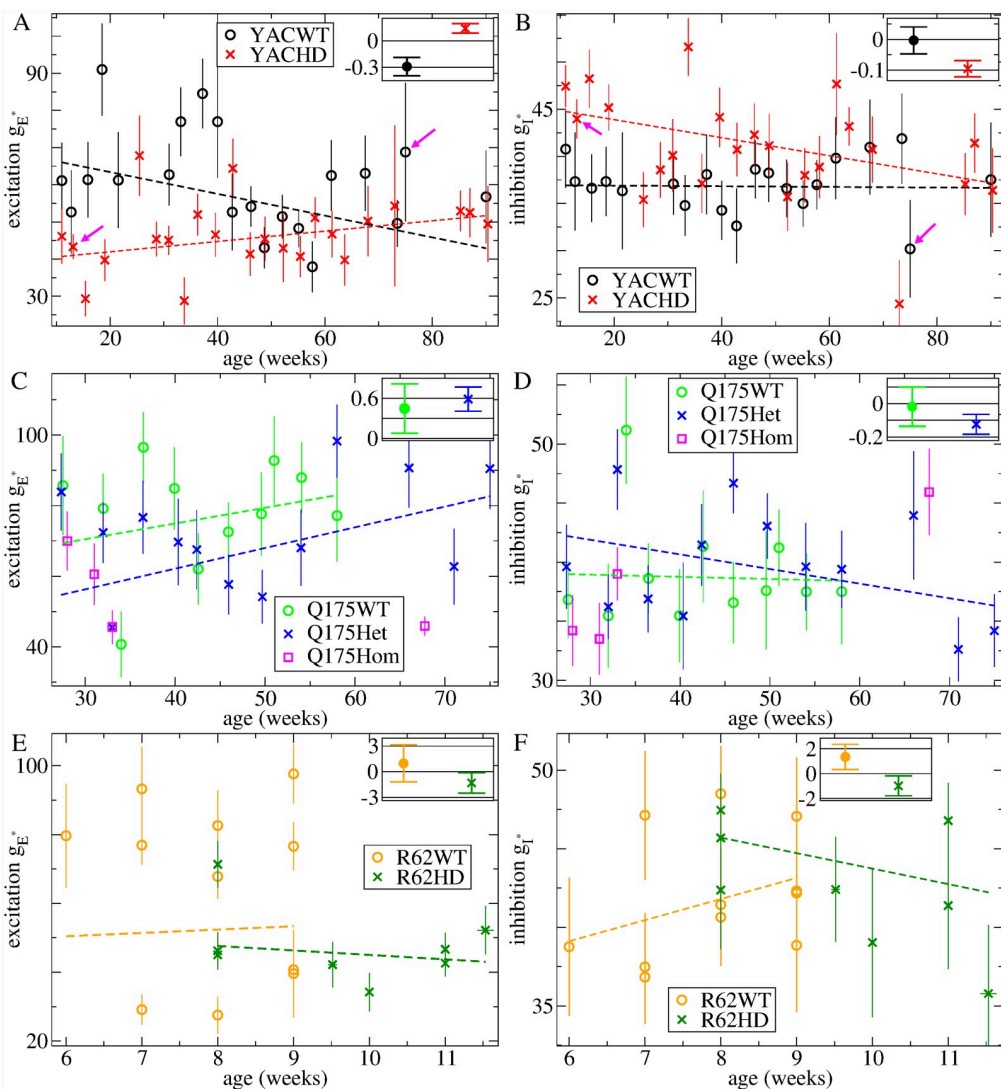

**Fig 11. Age dependency of best fit model excitation, $g_E^*$, (a,c,e), and inhibition, $g_I^*$, (b,d,f).** (a-d) Best fit model parameters estimated from experimental spike trains divided into three week interval datasets versus mean age of dataset. (a,b) YACWT (black circles), YACHD (red crosses), (c,d) Q175WT (green circles), Q175Het (blue crosses), Q175Hom (pink squares). (e-f) Model parameters estimated from experimental spike trains divided into datasets from individual animals versus mean age of dataset. R62WT (orange circles), R62HD (dark green crosses). (a-e) Bars show weighted standard errors in estimated quantities (see Methods). Lines show linear regression fits to YACWT (black), YACHD (red), Q175WT (green), Q175Het (blue), R62WT (orange), R62HD (dark green), plotted data, using errors in both x and y co-ordinates (see Methods). Insets show slopes and standard errors in the estimated slope (see Methods). p-values for the significance of difference of regression slope from zero calculated from two-tailed t-test are (a) YACWT, 0.011 (*), YACHD 0.017 (*), (b) YACWT 0.938, YACHD 0.001 (**), (c) Q175WT 0.25, Q175Het 0.006 (**), (d) Q175WT 0.879, Q175Het 0.057, (e) R62WT 0.660, R62HD 0.302, (f) R62WT 0.212, R62HD 0.252. (* denotes $p < 0.05$, ** denotes $p < 0.01$). (a,b) Pink arrows indicate the YAC datasets WT75 and HD12 whose heat maps are shown in Fig 6(h) (WT75) and Fig 6(i) (HD12) and whose ISI spike characteristics are shown in Figs 8, 9(a), 9(c) and 9(e).

Recurrent inhibition, $g_I^*$, also showed age dependency. In particular we find inhibition is enhanced in young YACHD mice, Fig 11(b, red crosses), compared to young YACWT mice Fig 11(b, black circles) and significantly decays with age, Fig 11(b, inset, red cross), although fluctuations across age groups are large. On the other hand, YACWT inhibition appears tightly

regulated with age, Fig 11(b, black circles, inset black circle), around $g_I^* = 35 \sim 40$. Interestingly, we find very similar behaviour in Q175 animals. Except for one outlier around 35 weeks, Q175WT inhibition appears tightly regulated, again around $g_I^* = 35 \sim 40$, Fig 11(d, green circles, inset green circle), while Q175Het inhibition, Fig 11(d, blue crosses), decays with age with a similar slope, Fig 11(d, inset blue cross), to the YACHD data, albeit with much weaker significance.

Thus the transition we found in the YACHD and Q175Het experimental data, Fig 10, mainly translates into a gradual decay of recurrent inhibition in HD compared to WT, starting from a state of enhanced inhibition. On the other hand feedforward excitation is reduced from the outset in HD and continues to be reduced across all age ranges. The tightly regulated inhibition around $g_I^* = 35 \sim 40$ found in WT animals places them in the AR just above the transition regime, Fig 4, at all ages, while HD animals are found deeper in the AR.

Due to the rapidly progressing nature of the R6/2 phenotype our data did not include much age variation. However in Fig 11(e) and 11(f) we plot estimated $g_{I,E}^*$ for R62WT and R62HD individual animals versus the mean age of their spike trains. Again we find excitation, $g_E^*$, is generally reduced in R62HD, Fig 11(e, dark green crosses), compared to R62WT, Fig 11(e, orange circles), while fluctuations in $g_E^*$ across R62WT datasets, Fig 11(e, orange circles), are again large, as they were in the YACWT datasets, Fig 11(a, black circles). Recurrent inhibition, $g_I^*$, does not show much phenotype dependency Fig 11(f), and none of the linear regression slopes, Fig 11(e,f insets), are significantly different from zero.

## Dynamical complexity

Our experimental data set was limited to single unit recordings. We did not have access to any multi-unit data and therefore did not use any multivariate information, such as cross-correlations, to estimate network model parameters. However, it is known that coherent burst firing and slowly varying cell-assembly activity is present in WT MSN multi-unit recordings and that this coherent activity is lost in HD mice [42]. Remarkably, this is also an emergent finding of the present analysis.

Fig 12 shows spike time series raster plots from the two network simulations best fit to the restricted age YAC datasets, WT75, and HD12, whose ISI statistics are shown in Fig 8, indicated by the arrows in Fig 4(a)–4(c). Evidently the WT best fit model with $g_I = 33$, $g_E = 50$ displays coherent bursting cell assembly activity, Fig 12(a), while this activity is lost in the HD best fit model with $g_I = 41$ and $g_E = 40$, Fig 12(b), despite the rather small change in parameters. These raster plots are very similar in appearance to the ones shown in [42] for WT and HD mice.

To quantify this finding, we calculate the principal components of the network activity based on the firing rate covariance matrix using a 100 ms sliding window to estimate firing rates from the spiking data. A large proportion of the variance is explained by fewer components in the WT model compared to the HD model, Fig 12(c). Fig 12(d) shows how the entropy of the explained variance distribution varies with $g_I$ and $g_E$ in multiple simulations. It has a minimum close to the transition from FR to AR at all levels of $g_E$, which is stronger for higher $g_E$. The minimum suggests that the network is generating low dimensional rate fluctuations in the transition regime. Entropy is high in the FR because although the mean-field input current dynamics has a stable fixed point, spiking generates white noise like fluctuations in the measured rate dynamics. On the other hand entropy is high at high $g_I$ because mean-field input currents fluctuate strongly. The squares indicate that the WT best fit model network (black) generates lower dimensional dynamics than the best fit HD model (red). The low dimensional rate activity shows up as coherent cell assemblies in the spiking activity, Fig 12(a).

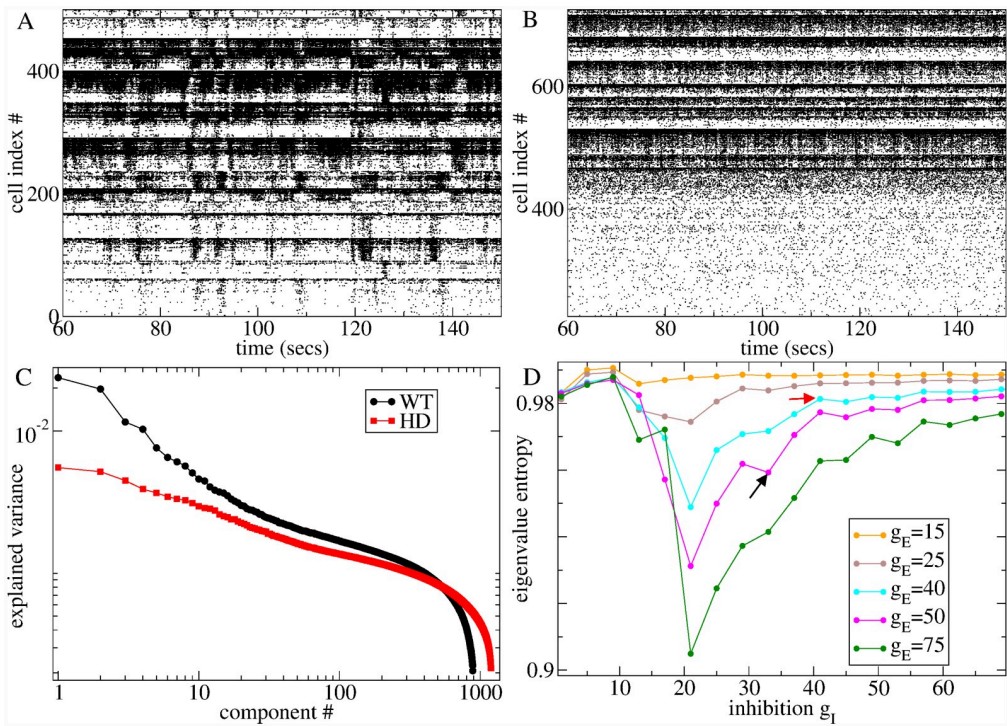

**Fig 12.** (a,b) Sections of raster plots from the (a) WT and (b) HD best fit network simulations depicted by crosses in Fig 6(f) (WT) and Fig 6(i) (HD) whose ISI statistics are also shown in Figs 8, 9(b), 9(d) and 9(f). (c) Explained variance of the principal components of ensemble rate fluctuations of all active cells for simulations shown in (a, black) and (b, red). (d) Entropy of explained variance distributions for principal components of ensemble rate fluctuations for multiple network simulations at different levels of inhibition $g_I$ and excitation $g_E$. Network simulations described in (a, b,c) are indicated by the black (WT) and red (HD) arrows.

## Discussion

MSNs in the striatum of WT mice display slowly varying coherent burst firing activity which is diminished in HD mice [42, 67, 68]. We hypothesized that this patterned activity is generated by the recurrent inhibitory MSN network and that its loss in HD occurs as a result of changes in parameters local to individual cells. To investigate this hypothesis we fit an MSN recurrent network model to mouse spiking data. To fit the model we varied only two parameters, the mean level of feedforward excitation and the mean level of recurrent inhibition. Other parameters, synaptic and cellular, were set at their physiological values and we used a fairly detailed cell model which faithfully represents MSN properties [79, 80]. In particular we demonstrate that it is neither necessary to vary any model timescale parameters nor is it necessary to rely on putative temporally varying cortical driving to reproduce slowly varying striatal burst firing activity or its reduction in HD. Given this simple model and its simplest possible parameter variation, the fits we obtain to both the WT and HD, as demonstrated by Figs 8 and 9, are surprisingly good. These fits are not just qualitative. Detailed properties of the autocorrelation function, as represented by the shape of the power-spectrum, and its decomposition into ISI distribution and serial ISI autocorrelation, are quantitatively captured. Fluctuations generated by recurrent inhibition in the MSN network are therefore sufficient to generate the correct ISI autocorrelation properties. That the WT can be so well represented and the HD phenotype recovered by just small parameter changes suggests that our hypothesis is correct.

By demonstrating the broad range of activity the recurrent inhibitory network model can generate as parameters are varied, Figs 4 and 5, we showed coherent bursting emerged as a property of a transition regime from stable to strongly fluctuating recurrent network dynamics. Most interestingly, we found WT empirical data was best fit by network models with levels of recurrent inhibition which placed them in the critical regime, just above this transition. This finding was consistent across the WT strains as shown by the KL distance plots, Fig 6(a), 6(d) and 6(g), and best fit values, Fig 7(b). It was also consistent across different ages in YACWT, Fig 11(b), and Q175WT mice, Fig 11(d), and across different individual R62WT animals, Fig 11(f). WT feedforward excitation, $g_E$, was more variable than recurrent inhibition, $g_I$, but also restricted to a given intermediate range.

Are the best fit values of inhibition and excitation we found physiologically plausible? The network simulation used to illustrate the fit to WT spiking data, Figs 6(f) and 8 (black), has recurrent inhibition level $g_I$ = 33. This is about 1/3 the value used in [80]. The discrepancy probably originates from the larger network size we are using which allows us to reduce the IPSP size closer to observed values. Indeed at this $g_I$ level, IPSPs generated in postsynaptic cells with membrane potentials close to firing threshold are about 2 mV (see Fig 3). This is in the physiological regime, but slightly larger than typically observed [10, 126, 127]. Typical values are around 0.5 ∼ 1 mV for cells close to firing threshold (IPSPs actually shown in [10, 126, 127] are much larger, but this is because the cells are voltage clamped and the chloride reversal potential is manipulated). The slightly larger size may be because we have fewer incoming inhibitory connections than in reality, which requires slightly larger IPSPs to provide the correct total recurrent inhibitory input to the postsynaptic cell. In our 2500 cell networks, with connection probabilites of 0.2, each cell receives connections from approximately 500 presynaptic cells. However experimental studies [10, 126] often find connection probabilities of around 0.35 (0.35 is the connection probability used in [80] model). Another possibility is that the IPSP decay timescale, although appropriate for GABA$_A$ synapses in the striatum, and close to the value used in [80], is a bit too small. Indeed while IPSPs are highly variable, IPSP areas shown in [126] often range between 100 and 300 $\mu V s$. Adjusting for the experimental protocol conditions results in areas between about 30 and 100 $\mu V s$ for IPSPs in cells close to firing threshold. The IPSPs shown in Fig 3 have areas around 60 or 70 $\mu V s$ well within the acceptable range, and close to the midpoint, in fact. Synaptic facilitation and suppression, which we have not included in this model, are also known to occur at these synapses [127], and would be expected to generate a larger range of IPSP sizes.

Furthermore, provided recurrent inhibition, $g_I$, has roughly the correct value, as we suggest it does, then feedforward excitation, $g_E$, should also have roughly the correct value if network activity is physiologically reasonable. As shown in Fig 8(a, solid black), the WT empirical mean ISI is around 2 secs. Since the best fit network generates an almost identical mean ISI, Fig 8(a, empty black), we can conclude that the level of feedforward excitation, $g_E$, we find is also reasonable. The fact that spiking characteristics matching WT data occur when the IPSP size, $g_I$, is physiologically reasonable, but, crucially, do not match when $g_I$ is outside this range, as demonstrated by Figs 4 and 5, futher supports our hypothesis that the empirically observed coherent bursting is generated by the MSN recurrent network.

We found that MSNs in the best fit WT model did not simply burst fire, but that their firing rates varied coherently across multiple cells, as shown by the spiking raster and eigenvalue spectrum, Fig 12. While the WT raster plot, Fig 12(a), does display switching activity resembling up-down state transitions in certain subsets of cells, we have not investigated if the membrane potential for these cells shows the bimodal distribution associated with up-down states in the striatum [70]. This coherent bursting activity was reduced in best fit HD models. Since our model estimation procedure utilized only single cell spiking characteristics, without

multivariate measures, this is an emergent finding. It is important to point out that the tacit assumption underlying this is that the model is correct. That is to say, the experimental data is actually drawn from the model with the network and cellular parameters we have set at their physiological values and within the explored range of $g_I$ and $g_E$. Given this, we have demonstrated in Methods that $g_I$ and $g_E$, can be uniquely determined from the single unit ISI statistics. Furthermore, network simulations show that at these parameter settings, coherent bursting is prevalent, Fig 12. Thus, we infer the presence of coherent activity from single unit ISI statistics by assuming the model is correct. This does not mean that this is the only model that could in principle reproduce the correct single cell ISI statistics or that they cannot be well reproduced at some other cellular or network parameter settings in the current model. In particular, it does not rule out the possibility that at some other parameter settings, single cell ISI statistics could be captured but without the cross-cell firing rate coherence. However, we have not observed such regimes in the current model, and they would have to be far outside the physiologically acceptable range. Furthermore, it is difficult to imagine a mechanism in the current model which could reproduce slowly varying burst firing, except by the coherent activity of afferent cells. Conversely, while we did not find it necessary to include a spatial dimension in the network structure, or network hubs, or to impose cortical driving activity which varied coherently across cells to reproduce coherent bursting, this does not mean that such mechanisms do not also operate in the striatum. However, it is highly likely that this is the simplest and most parsimonious model which accounts for coherent bursting in the striatum.

We investigated three different genetic models of Huntington's Disease. We found consistent changes in ISI statistics between the HD transgenics and the corresponding WT strains. In particular, low frequency spectral power and ISI CVs were reduced in the HD mice compared to WT mice, while the best fit ISI distribution, which was log-normal in WT mice, was much closer to gamma in HD mice. Due to the slowly progressing nature of the YAC and Q175 phenotypes we were also able to investigate changes in ISI statistics with age. Our YAC128 data ranged from 10 to 90 weeks, while our Q175 data ranged from about 30 to 75 weeks. In both cases we found that HD ISI statistics were most different from WT when young, but approached WT values with greater ages (to some extent). This was most evident in the HD best fit ISI distribution, which changed progressively from gamma to lognormal with age.

From the network model fit, we established that while the WT mice were poised in the network critical regime, strikingly, all three HD mouse types were found to be supercritical in the active regime. This change was mainly caused by a significant reduction in feedforward excitation, $g_E$, in all HD mouse types compared to their WT controls at all ages. Feedforward excitation did increase with age in YAC128 and Q175Het mice, but this effect was fairly weak. We also found changes in recurrent inhibition: $g_I$ was higher in young YAC128 mice, and to a lesser extent young Q175Het mice, than their age matched WT controls, and this was followed by a decline with increasing age. In WT mice inhibition, was much more tightly regulated, with no age dependency. R6/2 mice showed no age related changes however. Is there any empirical evidence for these model pathologies?

The large number of neuronal and glial dysfunctions associated with HD in the striatum complicate conclusions which can be drawn about the underlying mechanisms for dysregulation of burst firing. The two parameters we varied, $g_E$ and $g_I$, are simplifications of a much more complex reality. In our model $g_I$ controls the impact of slowly varying inhibitory input current from the MSN network on individual cells, while $g_E$ is a constant offset current which must be net excitatory. Many physiogically measured quantities that are pathological in HD can actually affect both of these parameters. Moreover, variations in both of these parameters affect the properties of the MSN network-generated fluctuations, as described in Fig 4, in a

non-linear way, such that it need not be immediately obvious which of them may underlie any particular empirically observed variation. For example, increasing $g_E$ increases the excitatory input to a cell and its firing rate, Fig 4(a). However, whether it increases or decreases the size or timescale of inhibitory input current fluctuations coming from the MSN network, resulting in an increase or a decrease of burstiness, CV, depends on the value of $g_I$, Fig 4(b).

HD pathologies, which may be represented by variations in $g_E$ in this model, include not only changes in presynaptic glutamatergic cortical and thalamic input and changes in feedforward inhibition, like FSI GABA input, but also alterations in glia or neuromodulators, like dopamine and endocannabinoids, which regulate the effects of these neurotransmitters on MSNs. Changes in postsynaptic dendritic structure or its intergrative properties and even MSN cellular changes, which may change rheobase current may also account for our variation in $g_E$. Similarly, changes in neuromodulators, glia, cellular or synaptic properties may also produce changes in $g_I$ if they alter synaptic transmission between MSNs. Furthermore, in this work we have not varied the network size or the MSN-MSN connection probability. However if the network is sufficiently large (which it is here) a change in the quantity of inhibitory synapses may produce very similar effects on network dynamics as a change in their mean strength $g_I$ [78, 102, 119, 123, 124].

Multiple studies [100, 128] have found cortical dysfunction in HD, that lead to changes in feedforward excitation to the striatum. In R6/2 mice, alterations in the corticostriatal pathway preceed symptomatology and MSN cell death. Spontaneous glutamatergic EPSC frequency was similar in 3-4 week old HD mice compared to controls, but significantly reduced by 5-7 weeks, when behavioural symptoms appear, and severely reduced by 11-15 weeks [24]. Corticostriatal AMPA and NMDA receptor-mediated evoked responses were reduced in size in symptomatic mice [129, 130] and the density of excitatory synaptic contacts onto MSNs, as well as MSN spine size, were found to be reduced [130]. All the R6/2 HD mice in our study were in this late symptomatic stage between 8 and 11 week and our primary result of reduced $g_E$, Figs 11(e) and 7(a), is thus strongly consistent with these experimental findings.

Studies of YAC128 mice have focused on three age groups, around 1.5 months (6 weeks), around 6-7 months (28 weeks) and around 12 months (52 weeks). We do not have 6 week data. However, at both 28 and 52 weeks, we find feedforward excitation, $g_E$, is reduced in WT compared to HD, Figs 11(a) and 7(a). This is also in good agreement with multiple studies. Cortically evoked EPSCs are smaller in HD than WT YAC128 at 7 months [26, 27]. Evoked postsynaptic receptor mediated NMDA currents in presymptomatic YAC128 mice decrease with advanced disease progression [26]. Spontaneous EPSCs are reduced in frequency and decay slightly faster in YAC128 compared to WT at 6 and 12 months [1, 40]. Evoked EPSCs were also much smaller in symptomatic HD mice compared to WT in D1 cells, although similar in D2 cells [1]. The authors therefore suggested that late stage HD is characterized by a loss of presynaptic and postsynaptic glutamate function in D1 cells in particular.

The 3-4 month and 6-9 month age groups of Q175 heterozygous and homozygous mice also demonstrate significant reduction in cortically evoked EPSC amplitudes compared to age matched WTs [98]. Although we do not have 3-4 month Q175 data, our analysis of 25-40 week data agrees with this. The same study also found that evoked EPSP amplitudes decreased progressively from Q175WT to Q175Het to Q175Hom in both age groups [98], in good agreement with our findings. Spontaneous EPSC frequency was also significantly decreased in MSNs from Q175 mice at 7 and 12 months and was more pronounced in Q175Hom than Q175Het mice [28]. This was associated with a reduction in spine density. Also in good agreement with our results, a recent study found dendritic excitability of indirect pathway MSNs in Q175Het mice was depressed at around 24 weeks when mice became symptomatic [131].

Interestingly, selective suppression of the mutant huntingtin gene in cortical efferents partially restores a healthy pattern of MSN activity in symptomatic HD mice [100], further supporting a role for aberrant cortico-striatal communication in HD.

Mechanisms responsible for decreased excitation may be further complicated by local homeostatic response by glia to changes in cortical inputs. GLT-1, the transporter responsible for regulating the extracellular level of striatal glutamate [132, 133] and for controlling glutamate-mediated long-term synaptic plasticity [134], is down-regulated in HD [135, 136]. Manipulations that increase striatal GLT-1 expression in HD mice improve both the motor phenotype [137] and striatal gamma band activity [138], however the decline in glutamate uptake was not reflected in an increase in extracellular glutamate [137]. The authors suggested that glutamate transmission in these mice may adapt to the loss of uptake with a compensatory decrease in glutamate release [137]. Downregulation of GLT-1 also increases astrocytic GABA release [139] which would contribute to the reduction in feedforward excitation, $g_E$. Our findings are therefore consistent with a decrease in glutamate release caused by downregulation in glutamate uptake and the concomittant increase in astrocytic GABA release. In addition, because GLT-1 is dysfunctional, AMPA receptor desensitization may also play a role in our findings of decreased excitation [140, 141].

Reduced feedforward excitation, $g_E$, may also reflect an increase in feedforward inhibition. The heterogeneous population of striatal interneurons exerts different inhibitory and modulatory control of MSN activity [142]. Growing evidence from HD model animals suggests changes in intra-striatal synaptic coupling related to interneuron dysfunction [23, 143]. Fast-spiking interneurons, for example, are likely drivers of the increase in striatal gamma-band power reported for Q175 mice [144] and this may also result in increased feedforward inhibition of MSNs.

Various MSN cellular neurodegenerative changes are also widespread in HD [86, 129, 145]. MSNs show increases in cell membrane input resistance, depolarized resting membrane potentials and reductions in rheobase current [40, 129, 146, 147]. The primary effect on cellular excitability is equivalent to an increased level of driving excitation in HD compared to WT. Thus the reduced feedforward excitation, $g_E$, we find in young HD compared to WT mice suggests that the well established reduction in synaptic efficacy plays a stronger role than the increase in cellular excitability at this age, which may represent a compensatory mechanism.

We also found a gradual decrease of excitation with age in YACWT mice while age related changes in excitation were not found in our R62WT or Q175WT mice, possibly due to greatly reduced age variation in our data for these latter two mice. While studies in mice appear to be lacking, in agreement with our model observation, decreased excitation in aged rats compared to young ones has been found [148–150].

While the reduction in cortical excitation in multiple transgenic HD mice is well established, experimental findings on recurrent inhibition are more complicated and less conclusive. Spontaneous and evoked IPSCs in R6/2 MSNs were found to be significantly larger and spontaneous ones more frequent than WT [10, 38]. These IPSCs may have originated from FSIs, so the authors also investigated MSN-MSN connections directly [10]. They found reduced connection probability between D2 cells and between D2 and D1 cells, but increased connection probability between D1 cells in R6/2 mice compared to controls. On the other hand success rates were higher in R6/2 mice in all connection types, while IPSP areas were smaller. A study of YAC128 mice found spontaneous IPSCs were more frequent, but had smaller area, in some MSNs compared to WT MSNs, and this difference increased with age between 6 and 12 months [40]. Spontaneous IPSC frequency was strongly increased in D2 but not D1 cells in symptomatic 12 month YAC128 mice [151]. In Q175 mice the frequency and amplitude of spontaneous IPSCs increased progressively from WT to Het to Hom at 7 and 12

months [28]. However, other studies [4] have found smaller evoked IPSCs in Q175 and R6/2 compared to WT mice.

Unfortunately most studies of IPSCs do not distinguish FSI feedforward inhibition from MSN feedback inhibition [86]. Increased feedforward inhibition is equivalent to reduced $g_E$ in our model. Furthermore, observed changes in the frequency of spontaneous IPSCs may be due to higher firing rates in presynaptic neurons [4] rather than any changes in synaptic strength *per se*. Progressive MSN cell death is well established however, and in some agreement with our results, would manifest as a gradual reduction in our parameter $g_I$ at advancing symptomatic ages, as would the general reduction in MSN-MSN connection probability found in [10]. Our model assumes a random network; asymmetric connectivity patterns [10, 152] and spatial distortions of dendritic trees [153–155] found in HD should be investigated in future modeling.

Both YAC128 and Q175Het mice were more similar to their WT controls when old than when when young. This was clear in both the raw experimental data and the fitted model. This finding is somewhat surprising because spiking activity might be expected to become more dissimilar between HD and WT as age and symptoms progress. However this need not be the case. In fact, it is possible that mutant mice start off in a pathological state, which normalizes with age through compensatory mechanisms, but which still results in behavioural symptoms. There are multiple possible scenarios consistent with this. Our data did not distinguish D1 and D2 type cells. One possibility is that reduced excitation, $g_E$, in young HD animals affects primarily one or other of these cell types, through abnormal corticostriatal synaptic structure, or dopamine or plasticity pathology for example [86]. As a result of this, these cells then gradually die or their connectivity changes [10] and in the process this reduces network inhibition, $g_I$. The weaker age dependent increase in excitation, $g_E$, could also occur through a number of mechanisms, for example related to glutamate reuptake mechansims or cellular excitability changes. Such processes could create an imbalance in D1 and D2 cell activity resulting in motor symptoms. Since D1 and D2 cells anchor the direct and indirect basal ganglia pathways, this might also produce the commonly observed biphasic HD motor symptom profile of hyperkinesia followed by hypokinesia at later stages [86].

We found WT model spiking displayed coherent bursting cell assembly dynamics while HD models did not, Fig 12. This was associated with a lower dimensional dynamical state created by the recurrent MSN network. Striatal cell assembly dynamics has been implicated in cognitive processing [43, 59, 60, 73, 74, 156, 157], including activation of particular cue-selective cell assemblies in primate response tasks [60]. Loss of appropriate switching between cell assemblies is found in disease pathology [42, 66, 67]. Dysregulation of coherent bursting was associated with motor symptoms in HD [42] while in Parkinsonian MSN networks, cell assemblies were found to be locked into a dominant state under dopamine depletion [66]. Therefore under pathological conditions, proper transitions between different cell assembly states that are required for locomotion may be prevented. It is possible that downstream structures in the BG [158, 159] utilize average signals from sequentially switching cell assemblies to control movement, which may be further divided into D1 and D2 dominant cell assemblies [156] for initiation and termination of sequence elements respectively. MSN activity clearly modulates movement parameters such as velocity and acceleration [160–162] such that any disruption of normal slowly varying patterned activity will have consequences for motor control.

The loss of coherently bursting cell assemblies is also closely related to the loss of low frequency power we observed in the HD power spectra. Our experimental power spectra show no sign of saturating up to the lowest frequency we investigated, Fig 2(a), 2(c) and 2(e). For a renewal spiking process, where individual ISIs are independently drawn from a given distribution, the power spectrum is fully determined by the ISI distribution. Renewal processes (with

finite variance) cannot generate diverging power spectra. This is possible only if ISIs have sufficiently slowly decaying serial autocorrelations [163–165]. Here the power-law diverging power spectrum, $S(f) \sim f^{-\beta}$, where $0 < \beta < 1$, is equivalent to a power-law decaying spike count autocorrelation function of the form $C(\tau) \sim \tau^{-\alpha}$, where $\alpha = 1 - \beta$ [163]. Stochastic processes with such autocorrelation functions are also known as 'long memory' processes because they decay much slower than the more typical exponentially decaying autocorrelation functions. Long timescales in MSN cell assembly dynamics have been observed [63–65] and a strong role for the striatum has been found in timing tasks. MSNs were found to activate sequentially in bursts across very long delay periods on the order of seconds [63, 64] and this activity pattern was stronger in the striatum than the cortex [65]. The reduction in $\beta$ we found may indicate a reduction in memory timescales in the striatum of HD animals, resulting in deficits in movement sequence planning on behavioural timescales.

Best fit WT models were found to be in a critical regime close to a dynamical transition. Network activity, in particular the quantity of burst firing, is highly responsive (i.e. susceptible) to small changes in cortical driving, $g_E$, in this regime, Fig 4(b). Thus WT networks may operate at a point where transfer of information from cortex through cortico-BG motor control loops occurs optimally. In contrast, although best fit HD networks were only a little further from the transition, variations in $g_E$ would have a much weaker effect on MSN network activity. This cortical-striatal disconnect may underlie the range of cognitive and motor symptoms found in HD.

Pharmacological interventions targeting early pathophysiological disturbances in HD mouse models can reverse neuronal dysfunction [14, 166] and delay progression to neurodegeneration [167]. Tetrabenazine (TBZ) has been found to provide significant benefit in the treatment of chorea associated with HD [168–170] while PDE10 inhibitors have also been invesigated [171]. The reduction in feedforward excitation, $g_E$, we found probably reflects changes in cortico-striatal synapses. It is possible that TBZ, by altering dopamine, normalizes corticostriatal transmission, and the effect of this could be tested in this network model in future work. Since recurrent inhibition, $g_I$, is also altered in the best fit HD model, it is not clear that striatal activity would be normalized, and coherent burst firing recovered, simply by agents that increase $g_E$ however. Such a strategy may even make matters worse. Moreover, it can be seen that the most appropriate pharmacological manipulations will depend on the stage of disease progression. In future studies, detailed models estimated from data could be used to test drug cocktails. This is particulary important in neural systems, since they are likely to exhibit criticality [172] and to be in the vicinity of dynamical regime transitions where complex feedbacks work non-linearly to complicate the effects of pharmacological manipulations, producing results not easily predicted from simple 'block and arrow' linear models. We hope that new insights provided by this modeling of the network and synaptic dysfunctions that take place in HD will stimulate further investigation of pathophysiological changes in this and other neurodegenerative disorders, and lead to the development of more effective drug combinations.

## Methods

### Animals

Data were obtained from three transgenic mouse models of HD and their corresponding wild-type (WT) controls: R6/2 (Bl6xCBA); YAC128 (FVB/N), and Q175 knock-in (C57Bl/6). Because robust neurological signs have a relatively early onset in R6/2s, this model was tested between 6 and 11 weeks of age. YAC128s and Q175s were tested beginning at 10 and 30 weeks of age, respectively, and continuing for several months over the course of symptom

development. The number of animals in each group was: YAC (WT) 30, YAC (HD) 35, R6/2 (WT) 11, R6/2 (HD) 8, Q175 (WT) 8, Q175 (Heterozygote) 15, Q175 (Homozygote) 4. All animals were housed in the departmental animal colony on a 12-h light/dark cycle (lights on at 07:30 h) with free access to food and water. All procedures followed the National Institutes of Health Guidelines for the Care and Use of Laboratory Animals and were approved by the Institutional Animal Care and Use Committee. All datasets were archival when shared with researchers from IBM, and no new experiments were suggested, designed, or performed based on the analyses reported here.

## Surgical procedures

Stereotaxic surgery for placement of electrode bundles in striatum followed established procedures for chronic electrophysiological recording (e.g., [42]). Briefly, mice were anesthetized with a mixture of chloral hydrate and sodium pentobarbital (chloropent, 0.4 ml/100 g, administered intraperitoneally) and were mounted in a stereotaxic frame. After an incision was made at the midline and the skull exposed, holes were drilled for bilateral recording in most cases, and multi-wire bundle electrodes were set in place (+0.5 mm anterior and ±1.6 mm lateral to bregma and 3.0 mm ventral to skull surface [173]. Two additional skull holes accommodated stainless steel anchor screws. Dental acrylic permanently attached the electrode assembly to the skull. After receiving standard postsurgical preparation, all mice were monitored for at least a week of recovery to ensure they were free from signs of pain and other health complications.

## Electrophysiological recording

Each electrode bundle was constructed in house and consisted of four, 25 or 50 $\mu$m Formvar-insulated stainless steel recording wires (California Fine Wire, Grover Beach, CA) and one 50 $\mu$m uninsulated stainless steel ground wire twisted together. Bundles were friction-fitted to gold pin connectors in a custom nylon hub (6-mm diameter). For recording, the electrode assembly was connected to a lightweight flexible-wire recording system equipped with field-effect transistors that provide unity-gain current amplification for each micro-wire. Neuronal discharges were acquired by a Multichannel Acquisition Processor, which allows for direct computer control of signal amplification, filtering, discrimination, and storage. To detect spiking activity, signals were bandpass filtered (154 Hz to 8.8 kHz) and digitized at a rate of 40 kHz. All spike sorting occurred before the animal was placed in the behavioural chamber for data collection. Sort Client software (Plexon) was used in conjunction with oscilloscope tracking to isolate each unit (matching the analog signal with the digitized template) and to eliminate the need for post hoc offline sorting. Voltage threshold >2.5-fold background noise was established, and a template waveform was created via principal component analysis. Autocorrelation and inter-spike interval analyses were applied to each unit to avoid recording the same unit on multiple channels. The recording system was connected to a swiveling commutator, which allowed the mice to behave freely.

Mice were placed in an open-field arena (26 x 18 cm) housed inside a sound-attenuating and electrically shielded recording chamber. Mice were behaviourally active during all open-field recording sessions. All mice participated in multiple recording sessions, typically at a rate of one session per week or every other week. Recorded units were treated as independent entities in each recording session because electrode drift and subtle changes in behavioural state cannot guarantee positive detection of the same neuron over multiple sessions [174]. All experiments were conducted during the light phase of the diurnal cycle. Individual recording sessions were 20-30 min in duration.

## Data analysis

We investigated MSN spiking activity from four HD mouse models (YAC128, R62, Q175Het, Q175Hom) together with the three corresponding strains of wild type mice (YACWT, R62WT, Q175WT). Spike trains were transformed to inter-spike-interval (ISI) sequences, denoted $I_i$, $i = \{1, .., N\}$. Some cells with anomolously high ISI skew >60, suggesting errors in the data record, were excluded from further analysis. Any cells with mean firing rates exceeding 10 Hz across the entire recording period were also excluded. This left: QWT 137, QHet 132, QHom 42, RWT 29, RHD 20, YACWT 210, YACHD 291 cells. Recording sessions had variable durations, $T_R$. These were: Q175 1200 secs, R62 3600 secs, YAC 1800 secs (a few YAC were 1200 secs). Model simulations (see below) always had duration 200 seconds. To make comparisons between ISI statistical quantities calculated from experimental data and model generated data, as well as between different mouse types, as direct as possible, all experimental spike trains were next divided into non-overlapping 200 second segments. Any such segments with less than 11 spikes were not used in analysis. Similarly, model generated single cell spike trains with less that 11 spikes were also not used in analysis. This is because calculation of ISI statistical quantities requires a minimum number of spikes. To utilize as much of the experimental data as possible and minimize bias towards cells with higher firing rates we only rejected segments with less than 11 spikes (10 ISIs), with the downside that such a small number can result in large finite size fluctuations in estimates of the statistical quantities. The resulting total number of 200 second spike trains in each of the seven mouse type datasets is described in Fig 13(caption).

Each mouse included recordings across multiple ages. To investigate age dependency in the data, the 200 second spike train segments were also subdivided into different age datasets. Data were subdivided into 23, 10 and 3 week age interval datasets for each of the seven mouse types. A given 200 second spike train with age $x$ would be included in all datasets where $\alpha i < x < \alpha(i + 1)$ and $\alpha \in \{23, 10, 3\}$ and $i \in \{0, 1, \ldots\}$ for its particular mouse type. The number of 200 second segments in each dataset that were actually used in the analysis reported in this

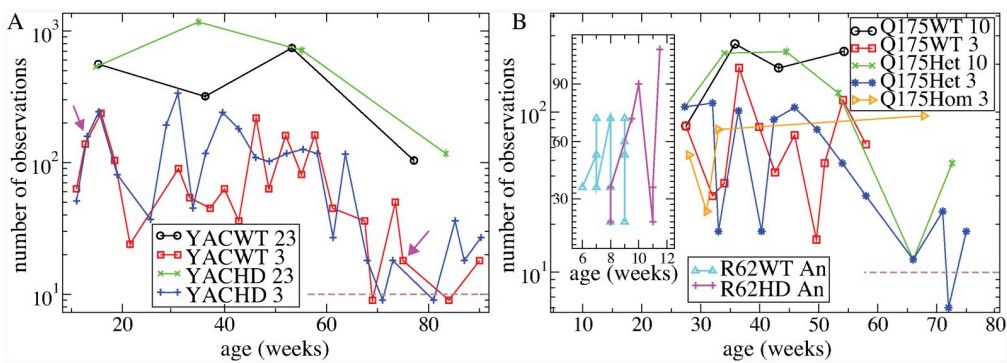

**Fig 13. Number of non-overlapping 200 second spike train segments versus dataset mean age for all datasets used in the analysis.** (a) YACWT and YACHD 23 and 3 week interval datasets (see key). Arrows indicate YAC WT75 dataset (including 18 observations with mean age 75 weeks) and YAC HD12 dataset (including 160 observations with mean age 13 weeks) whose ISI statistics are shown in Fig 8 and KL heatmaps in Fig 6(f) and 6(i). (b main panel) Q175WT and Q175Het 10 and 3 week and Q175Hom 3 week age interval datasets. (b inset) R62WT and R62HD individual animals. (a,b) Brown dashed line indicates datasets with less than 10 observations, which were not used in analysis. The total number of 200 second segments for the seven different full mouse type datasets were YACWT 1724, YACHD 2533, R62WT 526, R62HD 414, Q175WT 778, Q175Het 773, Q175Hom 251. Mean ages for the full mouse type datasets were YACWT 39.1097, YACHD 38.5703, R62WT 7.94275, R62HD 10.0078, Q175WT 42.3814, Q175Het 42.5106, Q175Hom 44.9569.

paper are plotted against the dataset average age in Fig 13. Only 23 and 3 week datasets were utilized for YAC mice, Fig 13(a), and only 10 and 3 week datasets were utilized for Q175Het and Q175WT mice, Fig 13(b). We only had enough Q175Hom mouse data to investigate 3 week interval (i.e. $\alpha = 3$) datasets, Fig 13(b), and we did not have enough data from R6/2 mice to subdivide them into different age datasets at all, however individual animals were investigated, Fig 13(b, inset). Any datasets with less than ten 200 second spike trains were not used in the analysis.

ISI statistical quantities shown in Figs 1, 8 and 10 are averages of the given quantity calculated for each of the 200 second spike train segments in the given dataset. The bars show the SEM over these calculated quantities for each given dataset.

Power spectra, $S(f)$, shown in Figs 2, 9 and 5 were calculated on each individual spike train in a given experimental dataset or model simulation dataset using a Morlet wavelet transform and then averaged across the spike trains in the dataset. More specifically spike count time series were created for each individual spike train by sliding a non-overlapping 4 ms window along the entire spike train and counting spikes in each window (almost always zero or one). Normalized Morlet wavelets $w(t, f)$ were constructed with frequencies starting from 1/64 Hz and increasing in multiples of $2^{1/8}$ Hz. The wavelets had temporal standard deviation $\sigma_t = 6/2\pi f$ and temporal widths $3\sigma_t$. Count time series were convolved with $w(t, f)$ to obtain complex time frequency representations $\text{TFR}(t, f)$ and the power spectra $S(f) = \langle |\text{TFR}(t, f)|^2 \rangle$ obtained where $\langle .. \rangle$ denotes the time average. Before averaging them, $S(f)$ were normalized by the mean firing rate of the particular spike train so that $S(f = \infty) = 1$. Figs 2, 9 and 5 show these normalized $S(f)$ averaged across all spike trains in an experimental dataset or simulation dataset to give dataset mean power spectra, where bars show SEM across observations in this dataset.

Cumulative ISI distributions $Q(n)$ shown in Figs 2, 9 and 5 were calculated from $Q(n) = \sum_{i=n}^{\infty} P(i) / \sum_{i=0}^{\infty} P(i)$ and $P(n) = \sum_{i=1}^{N} H(I_i - n\Delta\tau)H((n+1)\Delta\tau - I_i)(200/(200 - I_i))$, where the binsize $\Delta\tau = 10$ ms and the factor $(200/(200 - I_i))$ accounts for censorship of the ISI distribution by the recording period 200 secs. $Q(n)$ were calculated for each spike train in an experimental or simulation dataset, then averaged across the dataset, where bars show SEM in this dataset. In the same way serial ISI autocorrelations of lag $n$ shown in Figs 9 and 5 were calculated from $\rho(n) = (\langle I_{i+n}I_i \rangle - \mu)/\sigma^2$ and averaged across all spike trains in the particular experimental or simulation dataset, where bars show SEM in this dataset.

In order to demonstrate the size of fluctuations which can be expected in model network average mean quantites, model network simulation results shown in Fig 4 were calculated by dividing the 200 second simulations into 5 non-overlapping 40 second segments. For each 40 second segment, the given quantity was calculated for each active cell (those with more than 10 spikes in the 40 seconds) and then averaged across all active cells to generate network average quantities. The bars show the SEM across these five observations of network average quantities.

When estimating model parameters by comparing ISI quantities calculated from experimental data with the same quantities calculated from model generated data using the KL distance metric (see below) 200 second spike train segments were generated slightly differently. Instead of dividing a given experimental spike train of length $T_R$ into non-overlapping 200 second segments, we randomly drew a number, $10T_R/200$, of potentially overlapping 200 second segments to generate a larger number of samples. The resultant number of observations in each dataset was roughly ten times that shown in Fig 13. The quantity is not exactly ten fold because again any spike trains with less than 11 spikes were discarded. Groups with less than 100 spike train segments (which turned out to be the same as those with less than 10 non-overlapping segments shown in Fig 13) were not included in model fitting. Model generated data

always had length 200 seconds, and all single cell spikes trains with more 10 spikes were used in the calculation of KL distances.

From each of the obtained 200 second spike time series segments, experimental and model, the following statistical quantities are calculated.

- The firing rate $r$ (number of spikes per second in the 200 second interval).

- The mean ISI, $\mu = \langle I \rangle = (1/N) \sum_{i=1}^{N} I_i$, where $I_i$ is the $i^{th}$ ISI.

- The ISI coefficient of variation (CV), $\sigma/\mu$, where $\sigma^2 = \langle I^2 \rangle - \mu^2$ is the ISI variance.

- The rescaled ISI skew, S/CV, where the skew is S $= (\langle I^3 \rangle - 3\mu\sigma^2 - \mu^3)/\sigma^3$.

- The first two lagged ISI autocorrelations, $\rho(n) = (\langle I_{i+n}I_i \rangle - \mu)/\sigma^2$, where $n$ = 1, 2.

- Five intervals of the probability distribution of local CV which is bounded between 0 and 1. If $X_i = |I_{i+1} - I_i|/(I_{i+1} + I_i)$, where $|x|$ denotes the absolute value of $x$, these are defined by LCV(j + 1) $= \sum_{i=1}^{N-1} H(X_i - j/5)H((j+1)/5 - X_i)/(N-1)$, where $j$ = 0, 1, 2, 3, 4 and $H$ $(x) = 1$ if $x \geq 0$ and $H(x) = 0$ otherwise [103, 107].

- Maximum likelihood parameters (ML) for the log-normal distribution: scale $\mu_{LN} = \langle \ln(I) \rangle$; shape $\sigma_{LN} = \sqrt{\langle (\ln(I) - \mu_{LN})^2 \rangle}$.

- ML parameters for a gamma distribution: shape parameter $\sigma_\gamma = (3 - z + \sqrt{(3-z)^2 + 24z})/12z$, where $z = \ln(\mu) - \mu_{LN}$; log scale parameter $\ln(\mu_\gamma)$, where $\mu_\gamma = \mu/\sigma_\gamma$ [175].

- ML parameters for an inverse-Gaussian distribution: shape $\sigma_{IG} = (\langle I^{-1} \rangle - \mu^{-1})^{-1}$. (The scale parameter is $\mu$.)

To calculate KS distance from various ML distributions $D(n)$ we first obtain the censorhip corrected cumulative ISI distribution as described above, except now the binsize $\Delta\tau$ = 0.1 ms. Fits are performed using $Q(n)$ for $n$ = {1, ..., M}, and $M$ is the largest integer such that $Q(M) > 10^{-8}$. KS distance is the maximum value of $|1 - D(n) - Q(n)|$ over $n$, where $|x|$ denotes absolute value.

- KS Distance from ML gamma distribution: $D(n) = \gamma(\sigma_\gamma, n\Delta\tau/\mu_\gamma)/\Gamma(\sigma_\gamma)$, where $\Gamma(.)$ is the gamma function and $\gamma(.,.)$ is the incomplete gamma function.

- KS Distance from ML log normal distribution: $D(n) = 1/2(1 + \text{erf}((\ln(n\Delta\tau) - \mu_{LN})/\sqrt{2}\sigma_{LN}))$, where erf(.) is the error function.

- KS Distance from ML inverse-Gaussian distribution: $D(n) = \Phi(ax/\mu - a) + b\Phi(-ax/\mu - a)$, where $a = \sqrt{(\sigma_{IG}/n\Delta\tau)}$, $b = \exp(2\sigma_{IG}/\mu)$ and $\Phi(.)$ is the standard Gaussian cumulative distribution function.

Fifteen of these quantities, which we term 'features', are used to estimate the parameters of the network model. The features used were the $n$ = 1 and $n$ = 2 lagged autocorrelations, $\rho(n)$, the mean ISI, $\mu$, the ISI CV, the ISI rescaled skew, the firing rate, four of the quintiles of the local CV, LCV (1, 2, 4, 5), the two ML parameters from the gamma distribution, $\ln(\mu_\gamma)$, $\sigma_\gamma$, the two ML parameters from the lognormal distribution, $\mu_{LN}$, $\sigma_{LN}$, and the shape parameter from the inverse-Gaussian distribution, $\sigma_{IG}$. The KS distance measures were not used in the model fitting procedure. These features are far from independent. In fact some are simply functions and combinations of others. However since our estimation method is based on comparison of

feature distributions, this redundancy does not pose a problem. Futhermore the finite time length of our spike time series (model and experimental) affects different statistics in different ways. For example, $N$ spikes distributed over $T = 200$ seconds have a firing rate $r = N/T$, but the mean ISI may be far from $1/r$ if all the spikes occur in a small sub-interval of $T$. Furthermore, we noticed that some feature quantities appear to show much larger fluctuations across network simulations nearby in parameter space than others, in particular those involving higher moments such as the skew. Relatively large fluctuations in CV, Fig 4(b), across network simulations compared to much smaller fluctuations in the KS distances, Fig 4(d), and in the lognormal distribution shape parameter, $\sigma_{LN}$, Fig 4(c), can be seen in Fig 4. We found distribution maximum likelihood fit parameters were more stable than moments. The use of multiple different features in our model estimation procedure acts to reduce these fluctuations, to some extent.

Network simulation parameters are estimated from experimental data by comparing features calculated from the data, termed 'experimental features' with the same features calculated from the model, termed 'model features', using the KL distance of their joint distributions. More specifically, we choose a random set, R, of $n$ of the 15 features. For all the observations, $i = \{1, .., N_D\}$, in an experimental dataset, D, we calculate the experimental feature values, $F_{ij}^D$, for each of the features, $j = \{1, .., n\}$. For each feature, $j$, we find its median value, $F_j^D$, such that $\mathrm{Prob}(F_{ij}^D > F_j^D) = 1/2$. We compute the $n$–dimensional joint distribution, $P_{D,R} = \mathrm{Prob}(F_{i1}^D > F_1^D, F_{i2}^D > F_2^D, ..., F_{in}^D > F_n^D)$, over the $N_D$ observations, $i$. Similarly, for each observation, $k = \{1, .., N_M\}$, in a model dataset, M, we calculate the model feature values, $F_{kj}^M$. We obtain the $n$–dimensional joint distribution, $P_{M,R} = \mathrm{Prob}(F_{k1}^M > F_1^D, F_{k2}^M > F_2^D, ..., F_{kn}^M > F_n^D)$, over the $N_M$ observations, $k$. Finally we calculate the KL distance, $K_{M,D,R} = \Sigma P_{D,R} \ln (P_{D,R}/(P_{M,R} + \epsilon))$, where the sum runs over the $2^n$ bins and $\epsilon = 10^{-7}$ is a small number, in case $P_{M,R}$ contains bins with zero observations.

To determine best fit model parameters, $x_{D,R}$, to dataset, D, for the set of features, R, we first calculate model weights, $\alpha_{M,D,R} = K_{M,D,R}^{-\beta}/\sum_m K_{m,D,R}^{-\beta}$, which describes a normalized soft minimum over $K_{M,D,R}$, where the sum runs over all models $m$, and we used $\beta = 3$. $x_{D,R}$ is then obtained as the weighted average, $x_{D,R} = \Sigma_m x_m \alpha_{m,D,R}$, where $x_m$ is the parameter $x$ value for model $m$ and the standard error, $\sigma(x_{D,R})$, is given by $\sigma(x_{D,R})^2 = \Sigma_m (x_m - x_{D,R})^2 \alpha_{m,D,R}$.

This provides one observation of $x_{D,R}$ for the randomly chosen set of features R. To obtain a better estimate, the process is repeated for many (here 30) randomly chosen sets, R, of $n$ features from the full 15. For each set R we define $\phi_{D,R} = \sigma(x_{D,R})^{-1}/\Sigma_r \sigma(x_{D,r})^{-1}$ as a measure of goodness of fit. Finally we obtain the estimated parameter $x_D$ as the weighted average, $x_D = \Sigma_r x_{D,r}\phi_{D,r}$, with standard error, $\sigma(x_D)$, given by $\sigma(x_D)^2 = \Sigma_r (x_D - x_{D,r})^2 \phi_{D,r}$. These quantities, the weighted average and its weighted standard error, are the quantities shown in Figs 7 and 11 by the points and the error bars. Here we set $n = 7$, although $n = 5$ provides similar results, while results become noisy when $n$ is reduced further. Slopes shown in Fig 11 are obtained via standard linear regression of $x_D$ versus the mean age for observations in dataset D using $\sigma(x_D)$ for the error in $x_D$ and the standard error in the age in dataset D [176]. Error bars shown are the standard error in the slopes obtained from the linear regression in the usual way, [176] and p-values calculated using the two-tailed t-test on the slope normalized by its standard error with appropriate degrees of freedom are reported in the caption of Fig 11.

Fig 14 demonstrates this method for calculation of $x_D$, where experimental datasets, D, were generated from the network model simulations themselves. To make these results, each 200 second network simulation at given $g_E$ and $g_I$ is divided into two non-overlapping 100 second segments. The first set of segments are used as the experimental datasets, while the second set are used as the model datasets. Fig 14(a) shows that the estimated inhibition, $g_I^*$, is quite

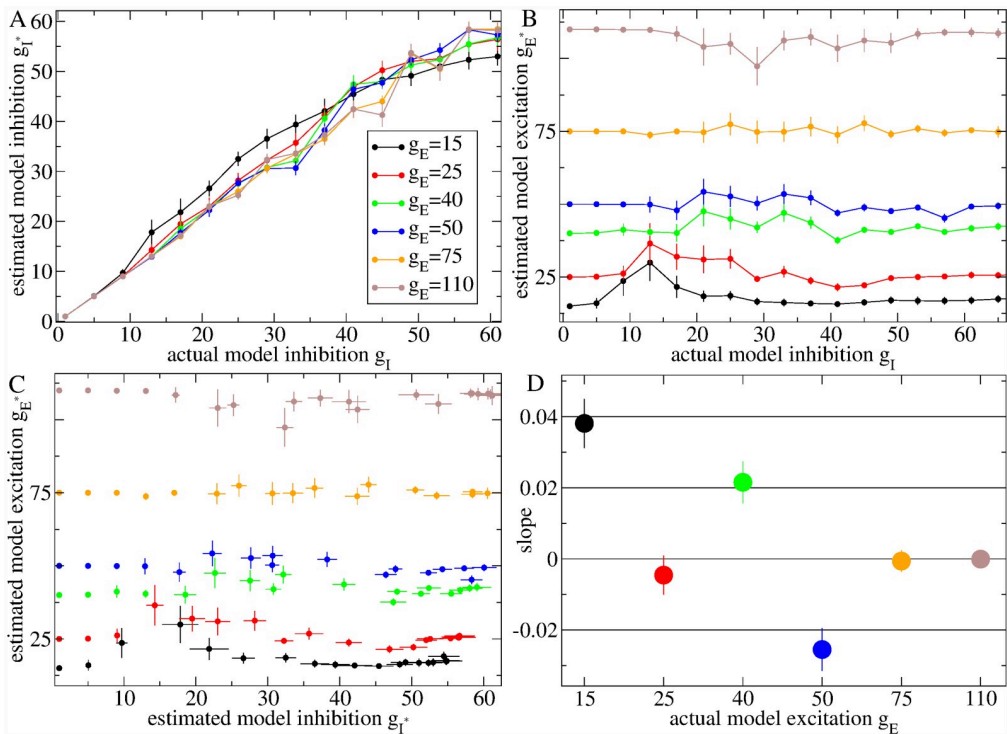

**Fig 14. Illustration of model parameter estimation method.** (a,b) Points show (a) estimated inhibition, $g_I^*$, and (b) estimated excitation, $g_E^*$, from data generated from models versus actual model inihibition, $g_I$, for various actual model excitation, $g_E$ (see key). (c) Points show the estimated excitation, $g_E^*$, versus estimated inhibition, $g_I^*$, for data generated from models. Colour (see key) shows the actual model excitation, $g_E$. Actual model inhibition, $g_I$, is not explicitly displayed the figure. (d) Slope of regression of estimated excitation, $g_E^*$, against estimated inhibition, $g_I^*$, for each level of actual model excitation, $g_E$. (a,b,c) Bars show weighted standard errors in estimated quantities (see Methods). (d) Bars show error in slope (see Methods).

close to the actual model recurrent inhibition, $g_I$, and independent of the model feedforward excitation, $g_E$. The method fails at large $g_I$ due to the presence of a maximum model inhibition, $g_I^{max}$. $g_I^*$ must always be smaller than $g_I^{max}$, producing a bias downwards at large $g_I$. Fig 14(b) shows that the estimated excitation, $g_E^*$, is quite close to the actual model excitation, $g_E$, and independent of the model recurrent inhibition, $g_I$. Again, at low $g_E = 15$, the estimated $g_E^*$ is biased upwards due to the absence of simulations with $g_E < 15$, and at high $g_E = 110$, the estimated $g_E^*$ is biased downwards due to the absence of simulations with $g_E > 110$. This effect is particularly strong in the critical model regime where fluctuations are largest. Fig 14(c) plots estimated $g_E^*$ versus estimated $g_I^*$ for all the datasets coloured according to their actual excitation, $g_E$. This demonstrates that these estimated quantities are independent without systematic bias. This is confirmed in Fig 14(d), where the slopes of regression of $g_E^*$ against $g_I^*$ for each $g_E$ are all close to zero.

Results in Fig 12 were calculated as follows. Rate time series were calculated from spike time series using a non-overlapping moving 100 msec bin. The cross-correlation matrix of these rates was calculated from all cells that fired at least 10 spikes in a simulation. Eigenvalues, $\lambda_i$, were calculated from the correlation matrix and their entropy as $-\sum_i p_i \ln p_i$ where $p_i = \lambda_i / \sum_j \lambda_j$. In raster plots, cells were organized using K-means clustering, with 30 clusters applied to the correlation matrix.

## Network model

We use the MSN cell model [79] exactly as in [80] including the modified leak current reversal potential of -90 mV based on more recent experimental findings [177]. All simulations include 2500 cells connected through inhibitory collaterals randomly with probability 0.2 [10, 126, 178–182]. The network degree distribution is therefore binomial. Each cell receives input from approximately 500 others with standard deviation of 20. Inhibitory synaptic currents are given by

$$I_{syn} = G_{syn}s(V - V_{Cl}),\tag{1}$$

where $V_{Cl} = -80$ mV is the chloride reversal potential. $G_{syn}$, the maximal synaptic conductance, is the parameter varied in network simulations governing the IPSP size. In a network simulation with given parameter $g_I$, connections are made with random strengths, $G_{syn}$, drawn from a uniform distribution on $[0.001g_I, 0.001g_I + 0.001]$. $s$ is a voltage dependent synaptic gating variable given by

$$\frac{ds}{dt} = aH(V)(1-s) - Bs.\tag{2}$$

We use parameters similar to those adopted in [80]: $a = 2$ and $B$ is a uniform random variable drawn from $[0.08, 0.09]$ for each presynaptic cell. [80] used $a = 2$ and $B = 0.1$. $H(V)$ is the Heaviside function applied to the membrane potential of the presynaptic cell. This is unity when the presynaptic cell spikes, and zero otherwise.

Feedforward excitation to MSNs is given by

$$I_{ex} = G_{ex}(V - V_{cat}),\tag{3}$$

where $V$ is the membrane potential of the postsynaptic MSN, and $V_{cat} = 0$ mV is the cation reversal potential. For a network simulation with excitation $g_E$, $G_{ex}$ is a random variable drawn uniformly from $[0.04381, 0.04381 + 0.002g_E]$ for each postsynaptic cell and held constant for the duration of the simulation. 0.04318 is the conductance at firing threshold. All cells in all simulations are driven above firing threshold by the feedforward excitation. Therefore all simulations are generated by varying only the two parameters $g_E$ and $g_I$. Simulations at different levels of $g_I$ were initialized with different random seeds, providing different realizations of the network structure and conductances $G_{syn}$ and $G_{ex}$. On the other hand, simulations at any given $g_I$ are identical, save for the variation in the parameter $g_E$.

The network model was implemented in IBM Model Graph Simulator, which is the core parallel processing architecture for model description (using the Model Description Language, MDL) and resource allocation (using the Graph Specification Language, GSL) of the Neural Tissue Simulator [183]. The Model Graph Simulator software is experimental. Readers are therefore encouraged to contact the authors if interested in using the tool.

## Ethics statement

All procedures followed the National Institutes of Health Guidelines for the Care and Use of Laboratory Animals and were approved by the Institutional Animal Care and Use Committee. All animal experiments were approved by the Bloomington Institutional Animal Care and Use Committee (approval numbers 13-002 and 16-001). All datasets were archival when shared with researchers from IBM, and no new experiments were suggested, designed, or performed based on the analyses reported here.

For histological verification of electrode placements, mice were deeply anesthetized with chloropent (at more than double the surgical dose) a current pulse ($30\mu A$ for 10 s) was passed

through each active micro-wire to mark recording sites. Mice were then transcardially perfused with saline followed by 10% potassium ferrocyanide $[K_4Fe(CN)_6]$ in 10% paraformaldehyde to produce small blue deposits at the site of the recording electrode ("Prussian blue" reaction). Brains were removed, postfixed in 10% paraformaldehyde for 1 h, and cryoprotected in 30% phosphate-buffered sucrose. The brains were then frozen; coronal sections ($60\mu M$) were then cut on a sliding microtome and mounted on gelatin-subbed slides. The sections were stained with cresyl violet and examined under a light microscope to confirm micro-wire location.

## Acknowledgments

We are grateful for the support of Casey Diekman and his lab at New Jersey Institute of Technology, where part of this research was conducted while AP was a visiting scholar.

## Author Contributions

**Conceptualization:** Adam Ponzi, George V. Rebec, James Kozloski.

**Data curation:** Scott J. Barton.

**Formal analysis:** Adam Ponzi.

**Funding acquisition:** George V. Rebec, James Kozloski.

**Investigation:** Adam Ponzi, Scott J. Barton, Kendra D. Bunner, Claudia Rangel-Barajas, Emily S. Zhang, Benjamin R. Miller.

**Methodology:** Adam Ponzi, George V. Rebec.

**Project administration:** George V. Rebec, James Kozloski.

**Resources:** James Kozloski.

**Software:** Adam Ponzi, James Kozloski.

**Supervision:** George V. Rebec, James Kozloski.

**Validation:** Adam Ponzi.

**Visualization:** Adam Ponzi.

**Writing – original draft:** Adam Ponzi.

**Writing – review & editing:** Adam Ponzi, George V. Rebec, James Kozloski.

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
