## [Decision Letter · Decision Letter 0]

27 Aug 2019

Dear Dr Ponzi,

Thank you very much for submitting your manuscript 'Striatal Network Modeling in Huntington's Disease' for review by PLOS Computational Biology. Your manuscript has been fully evaluated by the PLOS Computational Biology editorial team and in this case also by independent peer reviewers. The reviewers appreciated the attention to an important problem, but raised some substantial concerns about the manuscript as it currently stands. While your manuscript cannot be accepted in its present form, we are willing to consider a revised version in which the issues raised by the reviewers have been adequately addressed. We cannot, of course, promise publication at that time.

Sincerely,

Srdjan Ostojic

Guest Editor

PLOS Computational Biology

Kim Blackwell

Deputy Editor

PLOS Computational Biology

[LINK]

Reviewer's Responses to Questions

**Comments to the Authors:**

Reviewer #1: The present study by A. Ponzi and colleagues addresses the question of the pathophysiology of Huntington's Disease using theoretical modeling of network dynamics and experimental characterization based on single-unit recordings.

The authors first nicely introduce the topic and its challenges in their Introduction. The study then uses the properties of the interspike intervals (ISI) measured in vivo to constrain an existing network model of the spiny neurons assembly. They found different set of inhibitory and excitatory parameters when constraining from the Wild-Type and from the Huntington Disease genotypes. The main conclusion is that strongly coherent fluctuations explain WT network activity while HD activity is explained by a more constant network activity level with a non-trivial age-dependency phenomenon.

My personal opininion is that the ISI statistics is a noisy and very indirect measure to analyze the core phenomenon at work here (the fluctuations of network activity), and it doesn't seem to be an optimal approach to tackle this problem. The logic would have been more convincing if reversed. E.g. constrain the network model on population activity (either a low sampling of their SUA, or the LFP, ...) and then show that they can explain higher order quantities at the single neuron level (e.g. ISI statistics). But this is just a matter of opinion, so if the authors find that tackling this problem through the analysis of the ISI statistics is a good approach, this is fine with me.

In addition to this, I do have major concerns and minor comments, that I list below:

Major concerns:

1) How does the observed effect deviates from the experimental variability ?

There is no statistical analysis for the observations of line 152-165 and Figure 1 (also Fig. 9). You should plot errorbars on the summary data presented in Figure 1 to show how the observed effect compares to the variability across experiments/mice. The text in the legend: "Bars (exemplar results only for figure clarity) show SEM across the observations in the given datasets." is not clear. What is the "given datasets" here, it should be just the Q175WT dataset, right ? So why "datasets" in plural, is it averaged over datasets ? Just add the variabilty of each datasets in the plot with the N in the legend. Then, you can add panels to emphasize your point if needed.

2) No "N" in the paper. State the number of animals per genotype in the "Animals" section. I couldn't find the number of animals used, neither in the Methods nor in the Results.

3) Is the rather weak age-dependency statistically significant ? From the data of Figure 10, it doesn't seem obvious. I can not see any strong trend in Figure 10F. If this is a weak trend, tone down the importance of this phenomenon (from abstract to discussion).

4) The theoretical results should not depend on a specific realisation of the connectivity scheme. A state-of-the-art practice in the analysis of network dynamics in randomly connected assemblies is to run the simulations several times for different seeds generating the random connectivity. Then plot results as mean+/-sd in Figure 4,5,etc... And do the fitting on the mean values not on a single simulation case.

Minor comments:

- Figure 1: spreading the labels in all panels make the graphs poorly readable. Maybe make a legend at the top of the figure so that it's clear that it applies to all panels.

- line 191: dysregulation

- line 265: deterministic

- Abstract: I find that the term "dynamically critical" is misleading given what you study, different activity regimes in different conditions, but if you feel strongly about it, that's fine.

- Figure 13 has unreadable labels.

- The text is particularly long (also 14 Figures is quite a lot), in particular the discussion. I would advice to focus on discussing the most interesting observations (e.g. "age dependency of YAC and Q175-HD", line 877-896). But, that's a matter of taste again, so if the authors feel strongly about the whole text, that's fine with me.

- All figures. Maybe you could use a consistent-color code across figures if possible. That would make it easier for the reader.

Reviewer #2: The manuscript “Striatal Network Modeling in Huntington’s Disease” presents data and modeling results for three breeds of mice. Th authors quantitatively reproduced ISI distributions for WT and HD conditions by varying excitation and inhibition strength in the MSN network. The results suggest that the network is set at the transition between the frozen and active regimes, and this balance is violated in HD. The results are generally interesting. I find several points that need edits and clarifications. Thus, I suggest a minor revision.

Specific comments:

1. Lines 265-267 say that the model is entirely deterministic, but the next sentence says that fluctuations are generated intrinsically. Not sure how to understand this.

2. Fig. 4 is hard to understand because it shows only a dependence on the inhibition strength. I guess that the E values in the legend are actually the g_E, but it’s poorly described.

3. The authors compare the ISI distributions by calculating KS and ISI CV first (Fig. 2 and 4), but this analysis for the data and model are somehow separated. First they examine what distribution fits the experimental data best. Then they do the same for the distribution generated by the model. But in this part there is no direct comparison of the model and data distributions. I don’t understand why this is done this way.

4. Some lengthy explanations are hard to follow. For example, I couldn’t follow the long explanations to Fig. 5. There is no conclusion for each of this paragraphs. What are the results following from this data? In general, I would suggest emphasizing a conclusion to each paragraph.

5. There are no panel labels in fig 6. Probably because of that I couldn’t’ understand what the last two panels of this fig. present.

6. I couldn’t understand Fig. 13. What is the horizontal axis? It’s also cited right after fig 7, not in the numeric order.

**Have all data underlying the figures and results presented in the manuscript been provided?**

Reviewer #1: No: no link to the data is provided

Reviewer #2: Yes

PLOS authors have the option to publish the peer review history of their article (what does this mean?). If published, this will include your full peer review and any attached files.

Reviewer #1: No

Reviewer #2: No

---

## [Decision Letter · Decision Letter 1]

25 Nov 2019

Dear Dr Ponzi,

Thank you very much for submitting your manuscript, 'Striatal Network Modeling in Huntington's Disease', to PLOS Computational Biology. As with all papers submitted to the journal, yours was fully evaluated by the PLOS Computational Biology editorial team, and in this case, by independent peer reviewers. The reviewers appreciated the attention to an important topic but identified some aspects of the manuscript that should be improved.

We would therefore like to ask you to modify the manuscript according to the review recommendations before we can consider your manuscript for acceptance. Your revisions should address the specific points made by each reviewer and we encourage you to respond to particular issues Please note while forming your response, if your article is accepted, you may have the opportunity to make the peer review history publicly available. The record will include editor decision letters (with reviews) and your responses to reviewer comments. If eligible, we will contact you to opt in or out.raised.

- Supporting Information uploaded as separate files, titled 'Dataset', 'Figure', 'Table', 'Text', 'Protocol', 'Audio', or 'Video'.

We hope to receive your revised manuscript within the next 30 days. If you anticipate any delay in its return, we ask that you let us know the expected resubmission date by email at ploscompbiol@plos.org.

Sincerely,

Srdjan Ostojic

Guest Editor

PLOS Computational Biology

Kim Blackwell

Deputy Editor

PLOS Computational Biology

[LINK]

Please address the remaining concerns of Reviewer 1:

- Fig.4: provide error bars by bootstrapping either on neurons in the simulated network or time in the simulation

- Fig. 11: indicate which regressions are statistically significant

- clarify in the Discussion the limitation of the ISI approach for inferring network states

Reviewer's Responses to Questions

**Comments to the Authors:**

Reviewer #1: The paper has slightly improved in this revision. My suggestions have been partially taken into account. The revised paper does not clarify all aspects however and some of my concerns still hold to my opinion. The fitting procedure at the core of the study is seemingly still rather unstable and noise-sensitive, so I still struggle to be very confident about its output. My remaining comments are below.

Concerning the need of varying the seed across simulations

> “because the results shown in Fig.4 vary smoothly with the parameter variations (gE and gI)”

That is subjective. I was asking for the seed variations exactly because I didn’t find it smooth enough and because your “optimal value” for the WT looks slightly like an outlier with respect to the trend of the curve.

Also this could help the stability of your inference procedure. There are huge variations in conductance estimate for closely related subjects (of neighboring weeks) in Figure 11, this suggests that you conductance inference is very sensitive to tiny variations in experimental data and therefore not particularly trustworthy. If that's not seed variation, it could be something else, but anything that would stabilize the inference would bring confidence in the method.

Concerning the statistical analysis of the reported findings

> “Although some of the regression slopes appear significant according to our calculations”

And you do not find worth it to share this information ?? As readers, we are looking for quantitative conclusions about the phenomenon studied. Despite my previous comment you still do not provide any answer. If it is significant we want to know it. If it is not (either because there is no trend or because the fitting it too unstable), we also want to know it !

Concerning the conceptual limitation of relying on single neuron spiking to infer network regimes

> “In short we demonstrate that tackling the problem using the ISI statistics has turned out to be a highly effective approach.”

I am still not convinced that you can make any claim on network dynamics by only looking at ISI statistics. Network activity fluctuations can be virtually independent from the ISI distributions, because they are crucially constrained by the network synchrony (see e.g. Brunel JCNS 2000 for sets of regimes where either regular or irregular ISI can lead to drastically different network regimes because of different network synchrony).

There is nothing in your minimization procedure that relies on an estimate of the synchrony, so I can not see convincing evidence that the network regimes displayed by the model are physiologically relevant. To support this impression, the WT network is said to display Up and Down States (line 574, N.B. no quantification), personally I see the alternation of opposite activations and suppressions in two subpopulations. Why should it be Up and Down states ? (The Vm in *all* neurons should display the same slow oscillation, that wouldn’t be the case here).

To my opinion this is still a major weakness of the approach, it was rather superficially treated in this revision. The minimum would have been to clearly state and highlight this important limitation in the discussion.

Reviewer #2: My comments are addressed satisfactory.

**Have all data underlying the figures and results presented in the manuscript been provided?**

Reviewer #1: None

Reviewer #2: Yes

PLOS authors have the option to publish the peer review history of their article (what does this mean?). If published, this will include your full peer review and any attached files.

Reviewer #1: No

Reviewer #2: No

---

## [Decision Letter · Decision Letter 2]

9 Jan 2020

Dear Dr Ponzi,

We are pleased to inform you that your manuscript 'Striatal Network Modeling in Huntington's Disease' has been provisionally accepted for publication in PLOS Computational Biology.

In the meantime, please log into Editorial Manager at https://www.editorialmanager.com/pcompbiol/, click the "Update My Information" link at the top of the page, and update your user information to ensure an efficient production and billing process.

One of the goals of PLOS is to make science accessible to educators and the public. PLOS staff issue occasional press releases and make early versions of PLOS Computational Biology articles available to science writers and journalists. PLOS staff also collaborate with Communication and Public Information Offices and would be happy to work with the relevant people at your institution or funding agency. If your institution or funding agency is interested in promoting your findings, please ask them to coordinate their releases with PLOS (contact ploscompbiol@plos.org).

Thank you again for supporting Open Access publishing. We look forward to publishing your paper in PLOS Computational Biology.

Sincerely,

Srdjan Ostojic

Guest Editor

PLOS Computational Biology

Kim Blackwell

Deputy Editor

PLOS Computational Biology

Reviewer's Responses to Questions

**Comments to the Authors:**

Reviewer #1: The minor revisions requested by the editor were addressed.

**Have all data underlying the figures and results presented in the manuscript been provided?**

Reviewer #1: None

PLOS authors have the option to publish the peer review history of their article (what does this mean?). If published, this will include your full peer review and any attached files.

Reviewer #1: No

---

## [Editor Report · Acceptance letter]

8 Apr 2020

PCOMPBIOL-D-19-01096R2 

Striatal Network Modeling in Huntington's Disease

Dear Dr Ponzi,

I am pleased to inform you that your manuscript has been formally accepted for publication in PLOS Computational Biology. Your manuscript is now with our production department and you will be notified of the publication date in due course.

With kind regards,

Laura Mallard
